# Large size in aquatic tetrapods compensates for high drag caused by extreme body proportions

Susana Gutarra [1,2✉], Thomas L. Stubbs [1], Benjamin C. Moon [1], Colin Palmer [1] & Michael J. Benton [1]

Various Mesozoic marine reptile lineages evolved streamlined bodies and efficient lift-based swimming, as seen in modern aquatic mammals. Ichthyosaurs had low-drag bodies, akin to modern dolphins, but plesiosaurs were strikingly different, with long hydrofoil-like limbs and greatly variable neck and trunk proportions. Using computational fluid dynamics, we explore the effect of this extreme morphological variation. We find that, independently of their body fineness ratio, plesiosaurs produced more drag than ichthyosaurs and modern cetaceans of equal mass due to their large limbs, but these differences were not significant when body size was accounted for. Additionally, necks longer than twice the trunk length can substantially increase the cost of forward swimming, but this effect was cancelled out by the evolution of big trunks. Moreover, fast rates in the evolution of neck proportions in the long-necked elasmosaurs suggest that large trunks might have released the hydrodynamic constraints on necks thus allowing their extreme enlargement.

[1] School of Earth Sciences, University of Bristol, Life Sciences Building, 24 Tyndall Avenue, Bristol BS8 1TQ, UK. [2] Natural History Museum, Cromwell Road, London SW7 5BD, UK. ✉email: susana.gutarradiaz@bristol.ac.uk

Tetrapods have undergone multiple independent transitions from land to sea[1,2], each associated with major body plan innovations[1,3,4]. During the Mesozoic, numerous reptile clades took to the seas, becoming secondarily aquatic[5], of which the ichthyosaurs and sauropterygians, which emerged following the end-Permian mass extinction, were the longest-persisting and most diverse[6]. Derived members of both groups became pelagic, a shift in ecology coupled with the evolution of medium to large body sizes[7] (i.e. body lengths above 2 m), fast metabolic rates and the ability to thermoregulate[6,8–10]. Additionally, they became specialised, lift-based swimmers, but achieved this in different ways: ichthyosaurs as caudal oscillators, with stiff, deep bodies and broad lunate caudal tails[11,12]; and derived sauropterygians (i.e. plesiosaurs) as quadrupedal underwater fliers, with expanded girdles and large and rigid hydrofoil-like flippers[13,14]. The bodies of modern cetaceans are adapted for endurance and speed[15,16] and have often been used as a functional reference for ichthyosaurs[7,17,18]. However, the swimming mode of plesiosaurs is unique among tetrapods, and no living analogues possess similar body plans. Consequently, although in recent years the biomechanics of plesiosaur swimming has been the subject of extensive research[14,19,20], important aspects of their locomotory biology remain enigmatic.

Two distinct plesiosaurian body plans, the short-necked pliosauromorphs and the long-necked plesiosauromorphs, emerged independently in various clades[21]. The most extreme body proportions are found in the Cretaceous elasmosaurs, some of which had necks up to 6 m long[22,23]. Previous research suggested that ichthyosaurs could reach faster cruising speeds than plesiosaurs of the same body length[7,24], and that long-necked plesiosauromorphs were slower than the short-necked morphotypes[7,25]. These differences were based largely on assumptions of a less efficient swimming mode in plesiosaurs and, to a lesser extent, on differences in their fineness ratios (FR, the proportion of maximum body depth to total length). However, the effect of whole-body morphology on drag is not yet fully understood.

Certain streamlined axisymmetric geometries produce the lowest drag for a given volume at a fineness ratio of 4.5[26,27], which led to an assumption that FR = 4.5 is the optimal proportion for low drag in aquatic animals[7,28]. Consequently, long-necked plesiosaurs and other elongated aquatic reptiles possessing suboptimal fineness ratios, have been classified as 'slow'[7,25]. However, recent computational fluid dynamics (CFD) analysis in ichthyosaurs showed that drag is not correlated with fineness ratio for bodies of equal volume or mass[29]. Therefore, here we question this association between FR and drag also for plesiosaurs. Long necks have also been argued to add extra viscous drag due to their large surface area as well to increase pressure drag[7,25,30]. A recent CFD-based study of plesiosaurs concluded that drag was not affected by neck length during forward motion[20]. However, measures of skin friction (i.e. viscous) and pressure drag, the two main components of drag in fully submerged swimming, were not assessed, nor was the impact of the neck on the balance of drag to body mass[20].

Here we take derived ichthyosaurs and plesiosaurs as paradigm models for high axial and appendicular locomotory specialisations in marine tetrapods and compare them to modern cetaceans using digital modelling and CFD. Our CFD protocol informs on skin friction and pressure drag components[29], essential when assessing the flow over slender bodies for which drag is mostly frictional[31,32]. This allows us to address several questions. Did the bodies of both groups reach a similar level of low-drag form? Did the limbs contribute differently to drag in these groups? How do they compare to modern cetaceans? And importantly, how does body size influence the effects of shape? Then, we focus on plesiosaurs, exploring differences between the two morphotypes and the

interplay between body size and body proportions for derived plesiosaurs, which show a great spread in fineness ratios due to high neck length plasticity[23]. Lastly, we explore the evolution of trunk length (used here as a proxy for body size) and neck proportions in Sauropterygia (i.e. plesiosaurs and their closest Triassic relatives) and analyse the effect of neck plasticity (i.e. the variability of neck to trunk ratio) on the drag-related costs of steady swimming, discussing functional and ecological implications.

## Results and discussion

**Drag coefficients of plesiosaurs, ichthyosaurs and modern cetaceans.** At equal Reynolds numbers (same body length and same flow velocity), the total drag coefficients of plesiosaurs ($C_d$) are higher than the estimated values for ichthyosaurs and modern cetaceans (Fig. 1a). The limbless bodies, however, display similar $C_d$ in all three groups and are even lower-than-average in the long-necked plesiosaurs, indicating that the limbs are responsible for the observed high $C_d$. The limbs of plesiosaurs contribute to more than 20% of their total drag coefficient: up to 32.2% in the basal *Meyerasaurus* and averaging 25% in derived plesiosaurs, with no major differences between plesiosaur morphotypes. In parvipelvian ichthyosaurs the contribution of the limbs to $C_d$ is 11.2–15.6%, compared to 8.7–14.3% in modern cetaceans. Some of the living taxa we include provide a functional reference for this analysis. Our computed drag coefficient for the bottlenose dolphin model ($C_d = 0.00413$ at $Re = 10^7$) for example, is consistent with the estimates from a gliding living dolphin[33] ($C_d = 0.0034$ at $Re = 9.1 \times 10^6$) and other static CFD simulations[34] ($C_d = 0.00413$ at $Re = 10^7$). It is worth noting that these values are, as expected, lower than estimates obtained from kinematic models, as motion is not accounted for[35]. In a former study, drag coefficients for a plesiosaur (*Cryptoclidus*), two ichthyosaurs and various cetaceans were obtained from rigid models in water tanks[36]. However, the pressure drag component ($C_p$) was likely overestimated due to the proximity of the models to the air–water interface, and thus are not directly comparable to ours.

In all models across the various clades, velocity plots display a stagnation point at the anterior tip of the model, a thin velocity gradient along the body corresponding to the boundary layer, an area of higher velocity around the greatest diameter and a low velocity wake behind the body, characteristic features of a fully developed external flow (Fig. 1b, Supplementary Fig. 1). The acceleration of flow results in areas of low pressure (Supplementary Fig. 2), while high pressure areas are observed where stagnation occurs. Our CFD methodology has been previously validated against experimental data from slender torpedo-like shapes[26] and has been shown to provide a reliable distribution of internal drag components[29] essential when dealing with stream-lined bodies[35]. In all our simulations, the proportion of frictional and pressure drag was consistent with the expected values for slender geometries[31]: most of the drag originated from skin friction with a minor pressure drag component (Supplementary Fig. 2). The relatively larger limbs of plesiosaurs (Supplementary Table 1) produce a small increase in skin friction (Supplementary Fig. 2a), but a large increase in the pressure drag coefficient (Supplementary Fig. 2b), indicating that the latter largely explains differences in total drag coefficient between the groups. These effects might be explained by the low local Reynolds number of the flippers (resulting from a small chord length) producing high local $C_d$ relative to the rest of the body[31], alongside interference drag (i.e. drag caused by the interaction of flow fields where limbs and body meet), which might be higher for larger flippers.

**Effect of body shape and body size on drag-related costs of steady swimming.** When comparing morphologies at the same

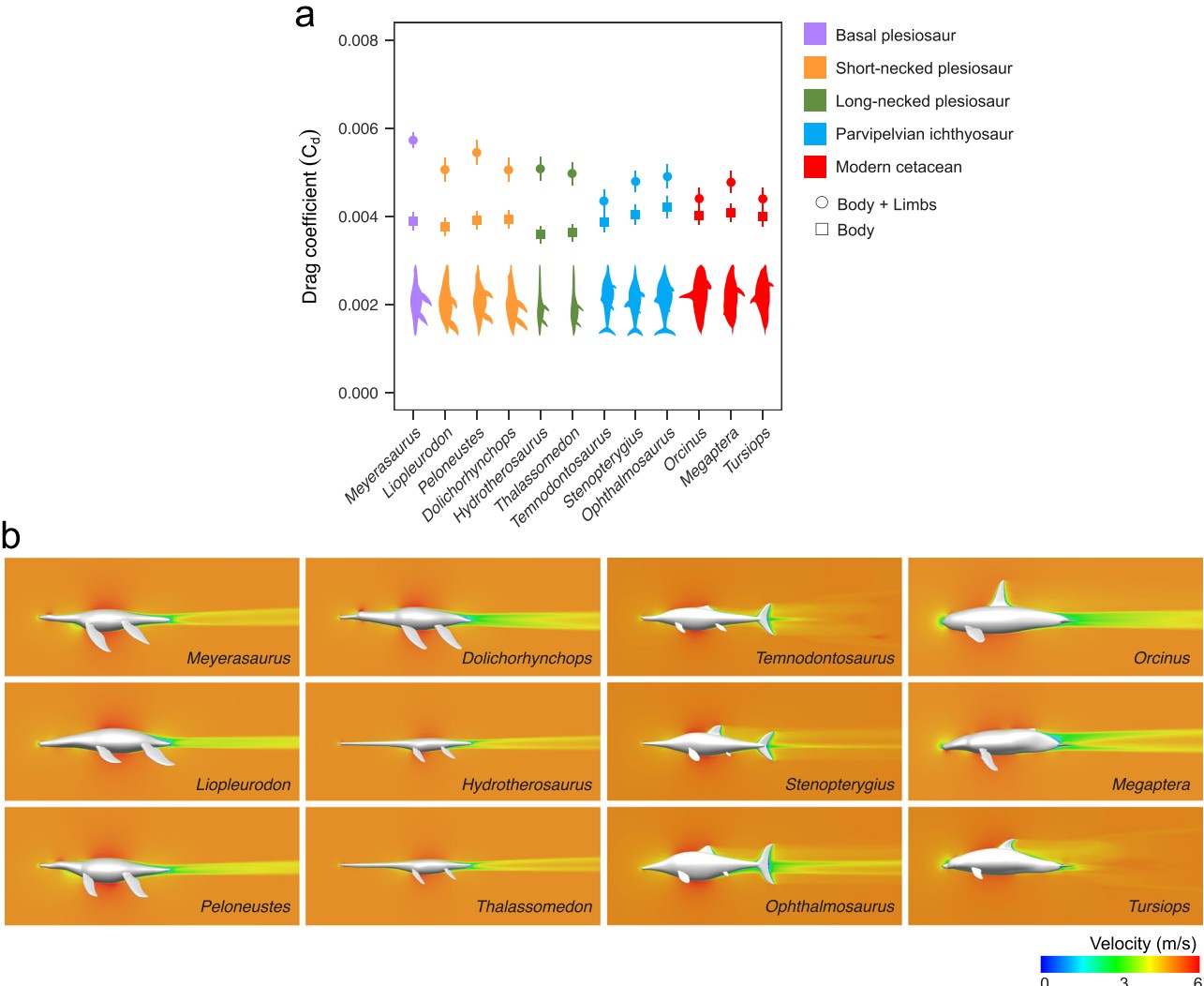

**Fig. 1 Comparison of the drag coefficient of derived plesiosaurs, ichthyosaurs and cetaceans. a** Total drag coefficient computed for the full models including the limbs ('body + limbs', circles) and the limbless models ('body', squares). Average (point) and range (bar) shown for calculations at Re = 5 × $10^6$–$10^7$. The derived short-necked plesiosaurs are highlighted in orange; the parvipelvian ichthyosaurs in blue and the extant cetaceans in red. A basal plesiosaur included as a reference is highlighted in purple. **b** Representative two-dimensional plots of the flow velocity magnitude at Re = 5 × $10^6$ (inlet velocity of 5 ms$^{-1}$) in lateral view. For dorsal view see Supplementary Fig. 1. Images of *Tursiops* and the three ichthyosaurs modified from Gutarra et al.[29].

volume (proxy for body mass) and the same velocity, to focus on the effect of shape alone, derived plesiosaurs produce on average 30% more drag than parvipelvian ichthyosaurs and modern cetaceans (Fig. 2a, Supplementary Table 3; two-sample *t*-tests $p < 0.001$). Drag-per-unit-volume represents the contribution of drag to the cost of transport ($COT_{drag}$), with $COT$ being the mass-normalised effort required for sustained forward swimming[37]. As for the drag coefficient, these differences are observed only when the full morphology is considered and not in the limbless models, indicating that the differences are caused by the relatively larger limb sizes in plesiosaurs. The model with the lowest absolute value of $COT_{drag}$ is *Tursiops*, against which all other taxa were normalised. The highest $COT_{drag}$ was estimated for the basal plesiosaur *Meyerasaurus*, which generates about 69% more drag than a bottlenose dolphin of the same mass. Among derived plesiosaurs, drag values increase from 29.2% in *Thalassomedon* to 42.6% in *Dolichorhynchops* relative to an equal-mass *Tursiops*, and no substantial differences are observed between the short-necked and long-necked morphotypes. The estimates of $COT_{drag}$ in parvipelvian ichthyosaurs are about 4% to 15% higher than for the *Tursiops* model, very close to our estimates for the modern

cetaceans *Orcinus* and *Megaptera*, which have relatively large fins. Overall differences between parvipelvian ichthyosaurs and cetaceans are non-significant (two-sample *t*-test $p = 0.63$; Supplementary Table 3).

Our CFD-based analysis thus shows that the overall morphology of plesiosaurs produced higher drag than parvipelvian ichthyosaurs and modern cetaceans, meaning that all other things being equal, an ichthyosaur should endure longer swims at a given speed or cruise at a faster velocity than a plesiosaur of the same mass. It is, however, uncertain to what extent all other things were equal. Propulsive efficiency estimates from living caudal oscillators such as cetaceans are generally higher than those of underwater fliers such as penguins, turtles and sea lions (for data of efficiency in extant animals and their sources see Fish, 2006[38]). However, plesiosaurs were quadrupedal swimmers, with no functional reference among living tetrapods, and recent work suggested that their propulsive efficiency was enhanced by fine-tuning of the fore and hind flippers[14]. Further, plesiosaurs have more surface area dedicated to producing thrust. Thus, whether a more efficient propulsion compensated the extra drag of the flippers in plesiosaurs is not yet known.

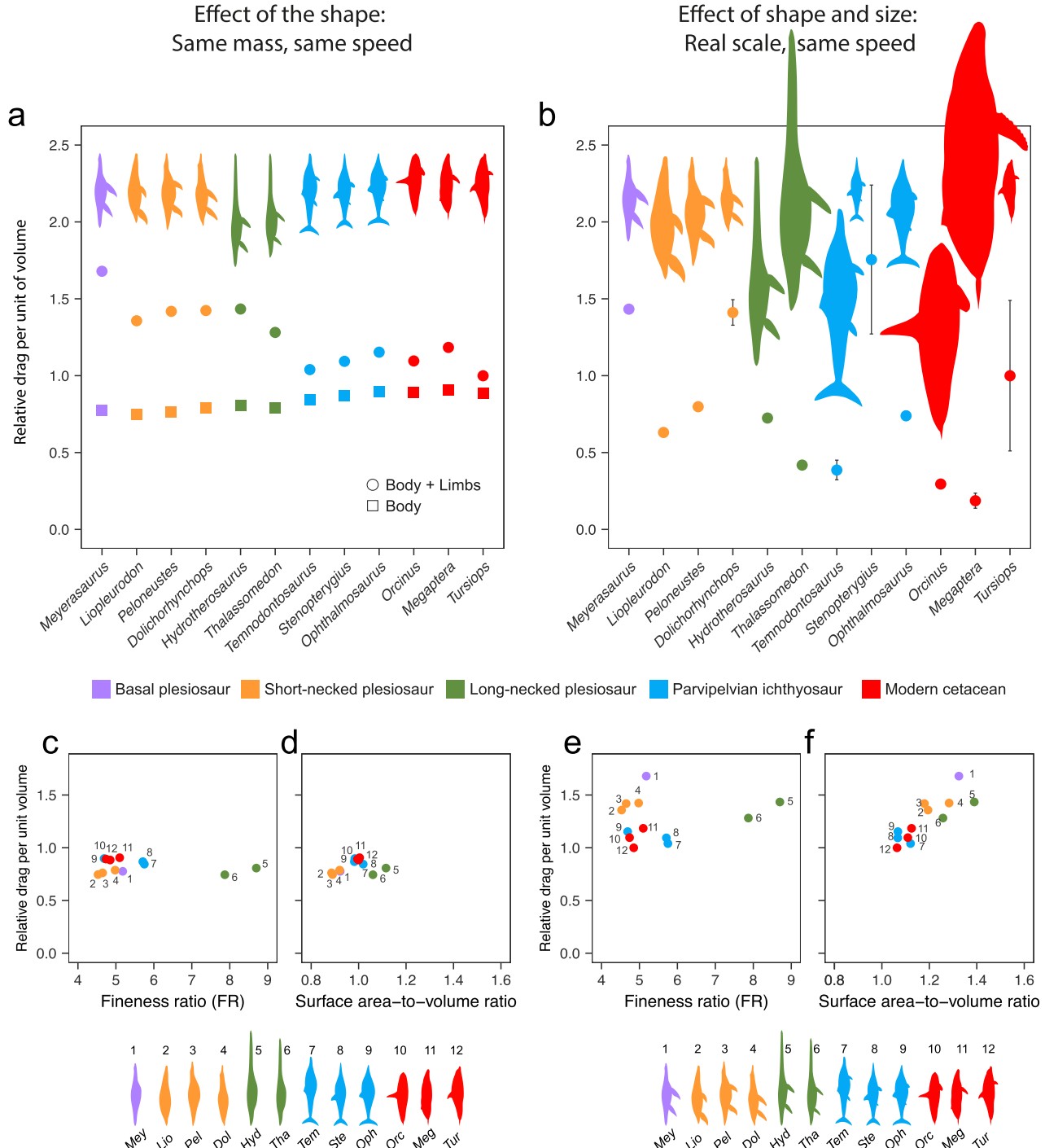

**Fig. 2 Effects of body shape and body size on the drag-related costs of steady locomotion for derived sauropterygians, ichthyosaurs and cetaceans.** **a** Relative drag per unit of volume (a proxy for the drag-related cost of steady locomotion or $COT_{drag}$) calculated for models scaled to the same total volume and compared at the same inlet velocity of $1\,ms^{-1}$. Results are shown for the full models including the limbs (circles) and the limbless models (squares). Average of calculations performed with two different volumes (see Supplementary Data). **b** Relative drag per unit of volume for life-size scaled models compared at the same inlet velocity of $1\,ms^{-1}$. Error bars represent minimum and maximum values accounting for taxon body size variation (see Supplementary Data). For an alternative set of calculations at $2\,ms^{-1}$, see Supplementary Fig. 3. **c–f** Relative values of drag per unit of volume for models scaled to the same volume and measured at the same inlet velocity of $1\,ms^{-1}$, corresponding to results in **a**, plotted against the fineness ratio, FR (**c**, **e**) and the surface area-to-volume ratio (**d**, **f**). Results are shown for limbless (**c**, **d**) and full (**e**, **f**) models. All values are normalised to the results for the *Tursiops* model. Derived short-necked plesiosaurs are highlighted in orange; the parvipelvian ichthyosaurs in blue and the extant cetaceans in red. A basal plesiosaur included as a reference is highlighted in purple.

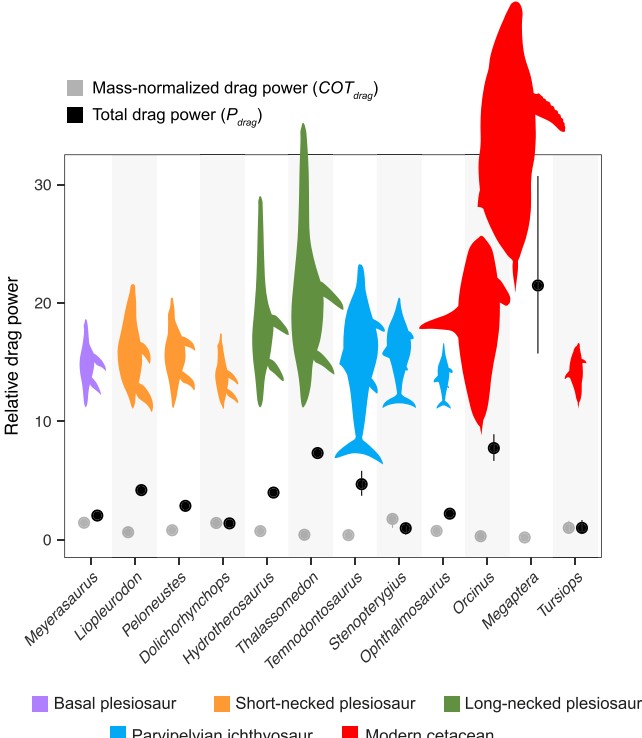

**Fig. 3 Comparative plot of mass-normalised drag power and total drag power.** Values of mass-normalised drag power (i.e., drag per unit of volume or $COT_{drag}$ calculated as in Fig. 2b) in grey, and non-mass-normalised total drag power, in black, for an array of derived plesiosaurs, parvipelvian ichthyosaurs and modern cetaceans compared at the same inlet velocity of $1\,\text{ms}^{-1}$. Error bars represent minimum and maximum values accounting for taxon body size variation (see Supplementary Data). Values are normalised to the results for *Tursiops*.

Our results show there is no correlation between the $COT_{drag}$ and the body fineness ratio (FR), regardless of whether the limbs are included or not (Fig. 2c, e). These observations, although opposite to what is generally assumed for aquatic animals[7], are consistent with previous analyses in ichthyosaurs[29]. The widely extended concept of a FR range for minimum drag comes from the study of aerodynamically engineered forms, and only applies to certain shapes when all other geometric parameters are kept constant[26,27], but cannot be extended to all complex streamlined forms. Instead, $COT_{drag}$ displays a strong positive correlation with the ratio of surface area to volume only in simulations with full morphology (Pearson's product-moment correlation, $r^2 = 0.89$, $p = 4.11 \times 10^{-9}$; Fig. 2f). This is consistent with the expected behaviour of flow over streamlined forms for which drag is mainly frictional[31]. The large hydrofoil-shaped limbs in plesiosaurs, necessary for their lift-based quadrupedal appendicular swimming[13,14,39], contribute to a large fraction of the surface area without adding much volume (Supplementary Table 1). In contrast, parvipelvian ichthyosaurs and modern cetaceans, both caudal oscillators, have lower proportions of body surface dedicated to the limbs (Supplementary Table 1), which add very little drag relative to a limbless body.

When body size is incorporated into the analysis (i.e. assessing the combined effect of shape and size by simulating the flow of life-size models for a constant velocity of $1\,\text{ms}^{-1}$), the group differences detected in the volume-scaled simulations disappear (Fig. 2b, Supplementary Table 3; all two-sample $t$-tests $p > 0.05$). In these conditions, the drag-related costs of steady swimming of plesiosaurs fall within the range observed in both modern

cetaceans and ichthyosaurs. Normalised against a 2.85 m-long *Tursiops*, the $COT_{drag}$ for derived plesiosaurs ranges from 0.42, estimated for the large elasmosaur *Thalassomedon*, to 1.41 in the medium-sized *Dolichorhynchops*. In the parvipelvians, $COT_{drag}$ spans from 0.33 estimated for the large *Temnodontosaurus*, to 1.76 in a 2.5 m-long *Stenopterygius*. Cetaceans show a smaller lower limit, because they include the largest animal in our sample, a 16 m-long humpback whale, with a $COT_{drag}$ of 0.13 compared to *Tursiops*. The estimated cetacean upper $COT_{drag}$ limit is 1.54 for a 1.9 m *Tursiops*. On the other hand, comparisons of the total drag power ($P_{drag}$, i.e., the non-mass normalised version of $COT_{drag}$) for the same speed of 1 ms$^{-1}$ (Fig. 3), show a different trend. $P_{drag}$ is highest for *Megaptera*, higher than in any fossil taxa included in this study, and is lowest in *Tursiops*. *Thalassomedon* is comparable both in total drag power and $COT_{drag}$ to the killer whale. Similarly, the thalassophonean pliosaurid *Liopleurodon* matches the elasmosaurian *Hydrotherosaurus* in having a similarly low mass-normalised $COT_{drag}$ but requiring about 4× more total drag power than *Tursiops*. Smaller forms like the polycotylid *Dolichorhynchops* and the thunnosaurian *Ophthalmosaurus* resemble the extant bottlenose dolphin in having a relatively high $COT_{drag}$ and low total power.

Thus, in contrast to the volume-normalised simulations, differences between animals at their life-size scale are mainly influenced by size. For example, medium-sized plesiosaurs and ichthyosaurs, such as *Dolichorhynchops* and *Ophthalmosaurus*, have values of $COT_{drag}$ close to that of a dolphin, while large plesiosaurs like *Thalassomedon* are more like the parvipelvian ichthyosaur *Temnodontosaurus* and a modern *Orcinus*. It is worth noting that the inflow velocity of 1 ms$^{-1}$, is a reference velocity used for comparative purposes, and is not equivalent to the optimal cruising speed (i.e. speed at which $COT$ is minimum[16]). This parameter is known to vary little in nature, with most vertebrates displaying values of preferred speed between 1–2 ms$^{-1}$ regardless of body size[40–42], which means it is reasonable to assume all tested taxa, regardless of their size, were able to swim at this velocity. Using a different reference velocity (2 ms$^{-1}$) has no effect on the relative values of drag per unit of volume and the mass-normalised drag power (Supplementary Fig. 3; Supplementary Data). A reduction of mass-normalised drag-related costs of cruising as body size increases is selectively advantageous, as energy savings can be used to extend foraging and mating range, increase swimming speed and fuel other activities[42,43].

Our analysis shows that for highly aquatic tetrapods, size dominates over shape in affecting the drag-related costs of steady locomotion. This is because $COT_{drag}$ (i.e., the balance of drag to volume) is highly sensitive to surface/volume proportion (Fig. 2f), and so is much influenced by isometry in streamlined animals.

**Interplay between neck anatomy and body size in plesiosaur drag.** Simulations at constant Reynolds number (i.e., comparing models at same total length and same flow velocity), show that necks up to 5× the length of the trunk do not increase substantially the total drag coefficient. Longer neck ratios up to 7× were found to impact the drag coefficient by as little as 3% (Fig. 4a). We estimated a 4–10% increase in skin friction drag coefficient for neck ratios of 3–7×, but also a comparable reduction in pressure drag resulting in almost no change in the total drag coefficient. A previous CFD-based study also found no differences in drag coefficient between plesiosaur models with variable neck proportions[20], but further comparison is not possible because of great differences in the order of magnitude of $C_d$, the use of a different scaling reference area and the lack of information on skin and pressure drag[20]. Here, we have shown

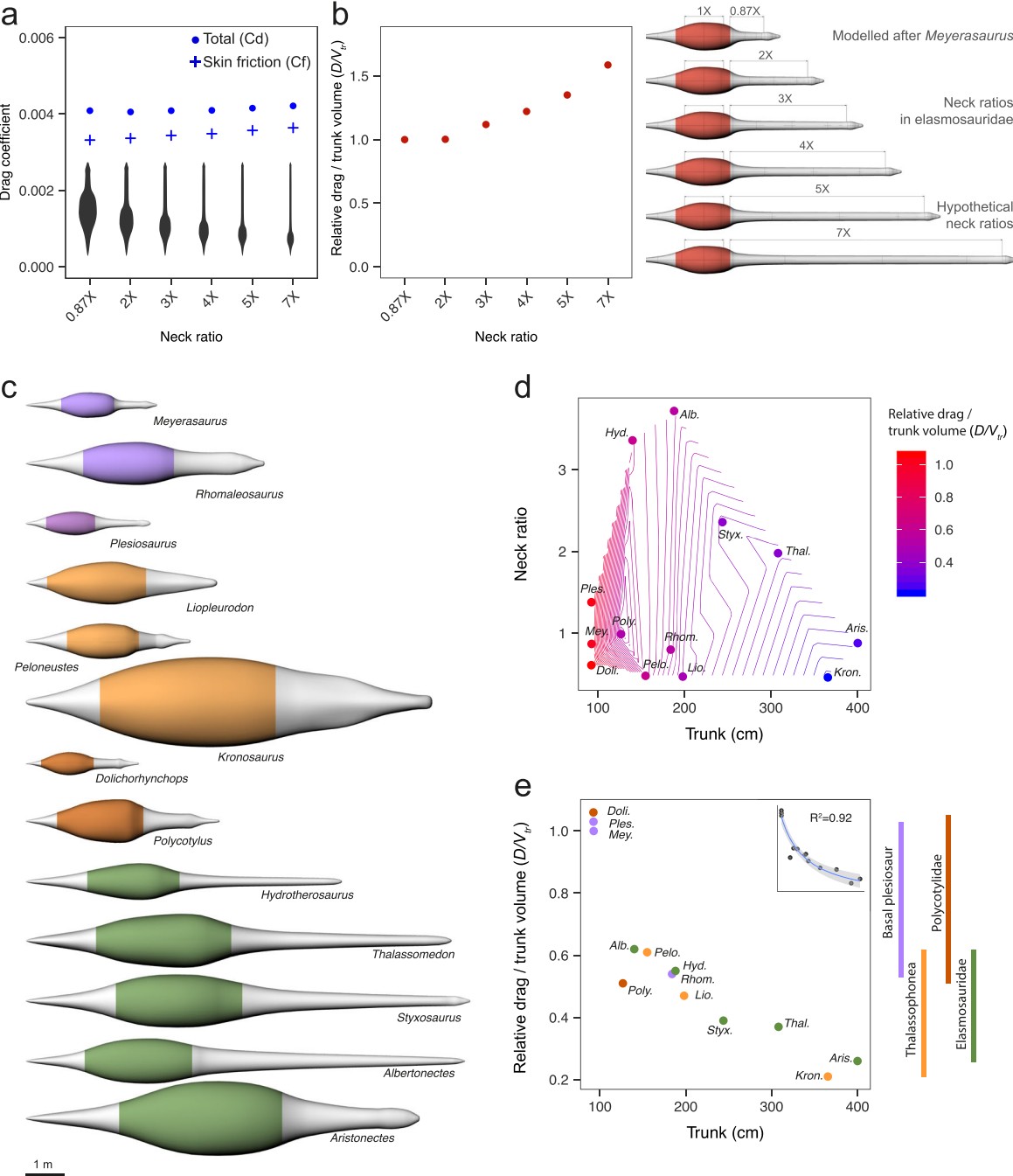

**Fig. 4 Influence of neck length and its interaction with body size on the drag-related costs of swimming in plesiosaurs. a** Total drag coefficient and skin friction drag coefficient for an array of hypothetical plesiosaurs with varying neck ratios computed at $Re = 5 \times 10^6$ (same total length and inflow velocity). **b** Drag per unit of trunk volume computed for the same array of models scaled at the same trunk length and tested at the same speed of $1\,ms^{-1}$. The hypothetical models were created by modifying the length in the model of the basal plesiosaur *Meyerasaurus victor* which has a neck ratio of 0.87×. The limits of the trunk (which extends along the torso and includes the edges of the pectoral and pelvic girdles) are shown in red in the rendered models. **c** Three-dimensional models of a wide array of plesiosaurs, in dorsal view, at their life-size dimensions, showing the differences in body proportions and sizes. The limits of the trunk in the models (defined as in **b**) are coloured by group. Basal plesiosaurs are highlighted in purple. Among the derived groups, thalassophonean plesiosaurs (derived pliosaurid plesiosaurs) are highlighted in light orange, polycotylid plesiosaurs in dark orange and elasmosaurid plesiosaurs in green. **d** Scatterplot of trunk length (cm) and neck ratio showing the relative drag per unit of trunk volume as a gradient of colour for each taxon analysed and for the plot area in between (contour lines represent the interpolated values of drag per unit of volume). **e** Plot of the relative drag per unit of trunk volume versus the trunk length showing results highlighted by group. Line plots at the right-hand side show the range for each group. The $D/V_{tr}$ and the trunk length show a significant negative correlation (Pearson's correlation coefficient calculated with log-transformed variables, $p = 2.28 \times 10^{-7}$, $R^2 = -0.92$). A small version of the fitted power curve (regression equation $y = 69.76x^{-0.94}$) is shown in the right upper corner. The grey area around the curve represents a confidence interval of 95%. All values in **b**, **d** and **e** are normalized to the results for the *Meyerasaurus* model.

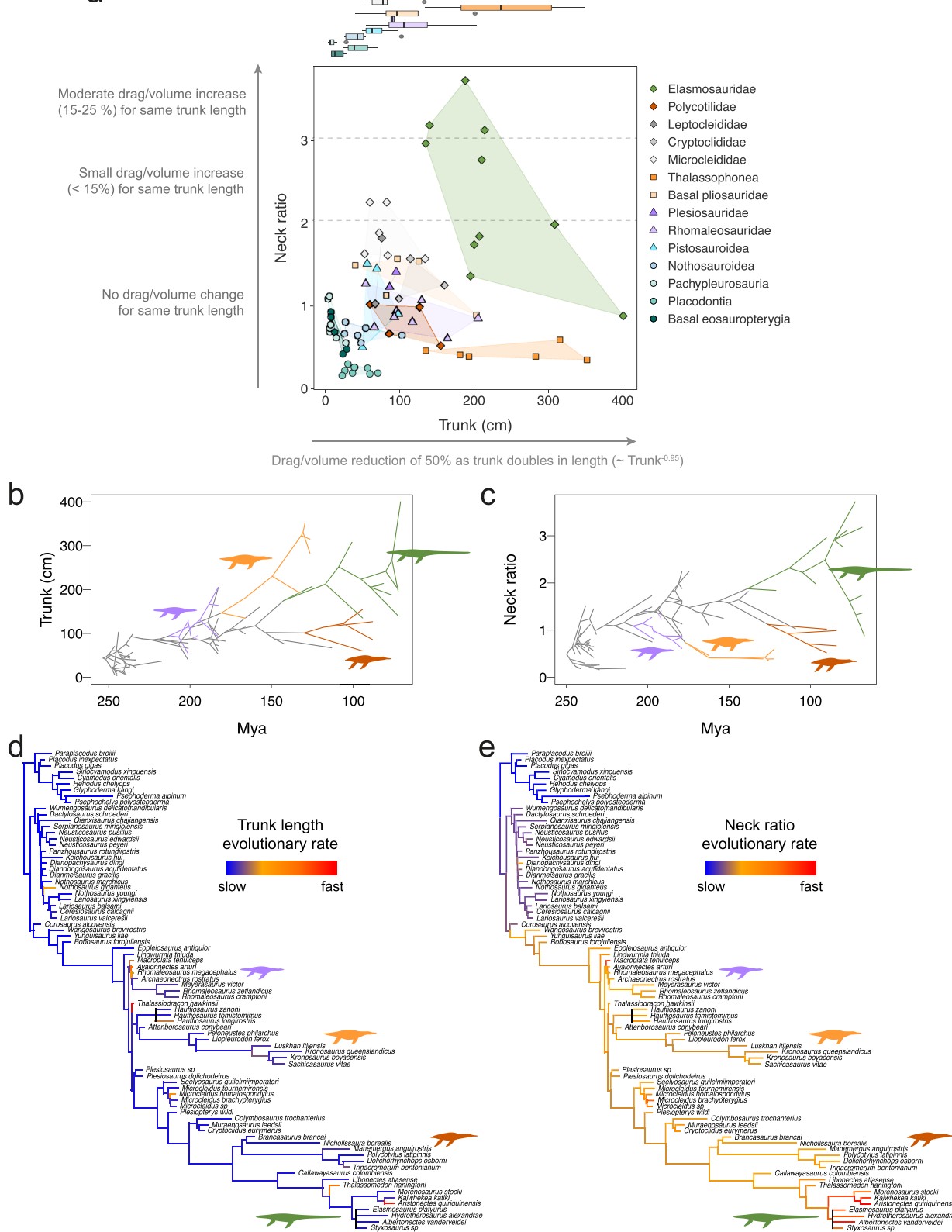

that long necks produce only a small increase in skin friction, although not as great as previously speculated[25,30], and this is nullified by reduced pressure drag.

Next, we explored the impact of neck proportions on drag-related costs of swimming in simulations where the size factor is removed. We found that if trunk dimensions are kept constant while the neck is enlarged, the drag per unit of trunk volume does not change appreciably for neck ratios up to 2×. However, longer neck proportions did impact resistive forces. This was moderate for a 3× ratio, with 12% more drag per unit of trunk volume, but became more substantial for longer necks, with 22%, 35% and 59% excess drag for necks of 4×, 5× and 7× respectively (Fig. 4b).

**Fig. 5 Evolutionary trends of neck proportions and body size in Sauropterygia and their implications for the drag-related costs of swimming. a** Bivariate plot of the length of trunk and the neck ratio of 79 sauropterygian taxa. Polygons in different colours show area occupied by the main sauropterygian groups. The functional trends describing the effect of each axis are based on results from flow simulations. On the top of this graph, a univariate plot shows the distribution and mean values of trunk length for each group. **b, c** Phenograms showing the disparity of trunk length (**b**) and neck ratio (**c**) in sauropterygians through time. The branches corresponding to basal Plesiosauria (including Rhomaleosauridae and Plesiosauridae), thalassophonean pliosaurs, polycotylids and elasmosaurs are highlighted (colour coding as in **a**). **d, e** Sauropterygian trees showing the evolutionary rates for trunk length (**d**) and neck ratio (**e**) represented by colour gradient (see Supplementary Fig. 5 for an alternative analysis to 5d using the log$_{10}$-transformed trunk length). Consensus trees show average results from analyses of 20 cal3-dated trees (see Supplementary Figs. 4 and 6 for analysis on Hedman-dated trees). Rates are based on the mean scalar evolutionary rate parameter.

This means that elasmosaurine elasmosaurs, with necks commonly 3–4× the length of the trunk[23] might have experienced higher drag than other plesiosaurs of similar trunk dimensions.

To test if the 'long neck effect' remains when body size is accounted for, we compared the relative amount of drag-per-unit-trunk-volume ($D/V_{tr}$) in a wide sample of plesiosaurs (Fig. 4c) at life-size scale for a constant velocity of $1\,ms^{-1}$, including three species with neck ratios above 2×: *Styxosaurus* (2.76×), *Hydrotherosaurus* (3.18×) and *Albertonectes* (3.72×), the last being the elasmosaur with the longest reported neck[44]. Our results show great variability in $D/V_{tr}$. Small-bodied plesiosaurs such as *Plesiosaurus*, *Meyerasaurus* and *Dolichorhynchops* generated up to six times more $D/V_{tr}$ than the largest plesiosaurs, *Kronosaurus* and *Aristonectes* (Fig. 4d, e). Comparisons per group show that both basal plesiosaurs and derived polycotylids, the groups with the smallest specimens, produced generally higher $D/V_{tr}$. Moreover, we did not find substantial differences between elasmosaurs and thalassophonean pliosauroids (Fig. 4e, Supplementary Table 4; all two-sample *t*-tests $p > 0.05$). Both groups had similarly low ranges of $D/V_{tr}$ regardless of neck length, lower on average than in polycotylids. These results stand even if we exclude *Aristonectes*, which belongs to the aristonectines, an elasmosaur subfamily with reduced neck length[23,45]. Further comparisons by morphotype show no significant differences between short-necked pliosauromorphs (here arbitrarily including plesiosaurs with neck ratios below 2×) and long-necked pliosauromorphs (Supplementary Table 4, all two-sample *t*-tests $p > 0.05$). The highest values of $D/V_{tr}$ occur in animals with trunk lengths of 100 cm or less, followed by a steep decrease between 100–150 cm and a steadier decrease in longer trunks. This indicates a strong negative correlation between trunk dimensions and $D/V_{tr}$ (Pearson's product-moment correlation between the log-transformed variables, adjusted $r^2 = -0.92$, $p = 2.28 \times 10^{-7}$). The curve that best describes this relationship is the power equation, $D/V_{tr} = 69.76 \times$ Trunk length$^{-0.944}$ (Fig. 4e), an almost inversely proportional relationship, consistent with the streamlined nature of these animals for which skin friction drag is dominant.

Polycotylids and thalassophonean pliosaurs, both derived pliosauromorph plesiosaurs[9,21], share the same general body proportions[9,21,46], but the latter had larger bodies and therefore needed less power in relation to their muscles to move at the same speed. Elasmosaurs on the other hand, despite their disparate morphologies, were no different from thalassophonean pliosaurs in their drag-related costs of forward swimming (Fig. 4c–e) and therefore they were likely to have been equally efficient cruisers.

Earlier research suggested that, even if long necks did not add extra drag during forward swimming, speed in elasmosaurs would have been limited to avoid added drag when their necks bent[20]. However, when the neck is bent in living forms, the course of swimming changes, as does the flow direction, but the body remains streamlined in the direction of incoming flow. For example, sea lions perform non-powered turns initiated by the head in which the body glides smoothly in a curved position,

limiting deceleration[47]. Further biomechanical research is needed to understand the role of plesiosaur necks in manoeuvrability and other aspects of swimming performance, as well as how these were influenced by shape and flexibility. The well-established idea that long-necked plesiosaurs were sluggish, slow swimmers[7,30] is thus not supported here, not because long necks did not increase drag[20], but because body size overrode this drag excess.

**Long necks evolved in large-bodied plesiosaurs: implications for drag.** We analysed trends of body size and neck proportion in a wider sample of sauropterygians, including plesiosaurian and non-plesiosaurian Triassic sauropterygians. Long necks (neck ratio > 3×) occur in taxa with trunk lengths > 150 cm, whereas most sauropterygians had neck ratios of ≤ 2× (Fig. 5a). The great plasticity of body proportions of sauropterygians before and after their transition to a pelagic lifestyle after the Triassic has been well documented[21,23,46], but this is the first time that neck and body size have been explored in the context of swimming performance for such a wide sample. We show that overall, sauropterygians and particularly plesiosaurs, mainly explored neck morphologies with little or no effect on drag costs and did not enter morphospaces that were suboptimal for aquatic locomotion (i.e., corresponding to small trunks with long necks; Fig. 5a). In fact, ancestral state reconstruction for trunk length shows that the ancestor of elasmosaurs was likely around 180 cm long and had a relatively short neck with a ratio smaller than 2× (Fig. 5b, c). This indicates that large trunks preceded neck elongation in elasmosaurs and suggests that extreme proportions might have been favoured by a release of hydrodynamic constraints.

We next explored evolutionary rates of relative neck length and trunk length in sauropterygians. The pattern of trunk length evolution is consistent with a heterogeneous rates model, not a homogeneous Brownian motion model (log Bayes Factor[48] (BF) > 5 in 100% of the sampled trees and > 10 in 92.5%, Supplementary Table 5). Analysis of non-transformed trunk data shows that through the evolution of Sauropterygia, there was a general increase in trunk length with some higher rates, in Triassic nothosauroids, Jurassic rhomaleosaurids and Cretaceous aristonectine elasmosaurs (Fig. 5d; Supplementary Fig. 4a). Additionally, analysis of the log$_{10}$-transformed trunk data highlights variation in the small-to-medium size ranges and reveals high rates in Triassic eosauropterygians (Supplementary Figs. 5 and 6). The largest trunks evolved independently in two groups, thalassophonean pliosaurids and elasmosaurid plesiosauroids, with no evidence of high rates in the former. In the plesiosauroids, rates are not particularly high in the basal branches, but they are very high in derived aristonectines, and rates for the whole clade were significantly higher than the background rate in 40% of randomisation tests (Supplementary Fig. 7 and Table 6). A progressive increase in body mass over evolutionary time has been described for various clades of aquatic mammals[49] and seems to be a common hallmark of the aquatic adaptation to marine pelagic lifestyles in secondarily aquatic

tetrapods[44]. Whether body size reaches a plateau as is the case in cetaceans[49] and what constraints influence the evolutionary patterns of size in plesiosaurs remains unexplored. Against this general trend, some derived plesiosaurs, such as polycotylids, saw a reduction in body size, which might have been related to pressures on niche selection, such as adaptation to specific prey, the need for higher manoeuvrability or other ecological factors. As shown earlier, small sizes require lower amounts of total power for a given speed, and therefore would be favoured if for example food resources were limited. This suggests that, in spite of the energy advantages of large size in terms of reduced mass-specific drag[29] and metabolic rates[49,50], which make it a common adaptation to the pelagic mode of life, other constraints limiting very large sizes were also at work[50,51].

A heterogeneous evolutionary rates model for neck proportion is also strongly supported (log BF > 5 in 100% of the sampled trees and > 10 in 45%, Supplementary Table 5). Fast rates are consistently seen at the base of Pistosauroidea (including some Triassic forms and plesiosaurs) and, interestingly, also within elasmosaurs (Fig. 5e; Supplementary Fig. 4b). The neck proportions of elasmosaurs were found to evolve at a faster pace than the background rate in 90% of analyses (randomisation test $p$-value < 0.001 in 80% and < 0.01 in 10% of the sampled trees; Supplementary Fig. 7 and Table 6). Very fast rates in elasmosaurs are concentrated in the most derived branches (i.e. Euelasmosauridia from the late Upper Cretaceous[52]) and represent both rapid neck elongation in elasmosaurines and rapid neck shortening in weddellonectians (i.e. aristonectines and closely related taxa[52]). Additionally, various other independent instances of relative shortening of the neck occurred during the evolution of Sauropterygia, most notably in placodonts, pliosaurs and polycotylids, but these are not associated with high rates.

Our findings contrast with a previous study[23] which did not identify any significant evolutionary rate shifts in the neck ratio across Sauropterygia. Here we use a larger number of taxa and a different model fitting approach, which might account for these discrepancies. The association between very long necks and large trunks, along with our flow simulations results and the evidence of high rates in the elongation of necks in elasmosaurines (Fig. 5e), suggests that neck elongation was facilitated by large body sizes. The question remains why neck ratios did not evolve longer than 4×. According to our data, hydrodynamic constraints might have operated against the selection of such long necks. However, it is possible that the primary function for which they were selected, which is still debated[30,53], did not require necks with those characteristics. Neck anatomy is likely to be the result of a compromise between different functions/constraints, one of them being hydrodynamic, as shown by the results presented herein.

## Methods

### 3D reconstruction of plesiosaurs, ichthyosaurs and modern cetaceans. 
Six very complete plesiosaur specimens were selected for reconstruction as full-body, three-dimensional models: one basal plesiosaur, (i) the rhomaleosaurid *Meyerasaurus victor* (specimen exposed in ventral view, SMNS 12478) from the Early Jurassic of Germany; three derived short-necked plesiosaurs, including two thalassophonean pliosaurids, (ii) *Peloneustes phylarchus* (3D mounted specimen GPIT-RE-3182, previously GPIT 1754/3) from the Middle Jurassic of Germany and (iii) *Liopleurodon ferox* (3D mounted specimen GPIT-RE-3184, previously GPIT 1754/2) from the Middle Jurassic of the UK, and (iv) the polycotylid *Dolichorhynchops osborni* (3D mounted specimen KUVP 1300) from the Late Cretaceous of North America; and two long-necked elasmosaurid plesiosaurs, (v) *Thalassomedon hanningtoni* (3D mounted specimen DMNH 1588) and (vi) *Hydrotherosaurus alexandrae* (UCMP 33912, figured and reconstructed by Welles[54]) from the Late Cretaceous of North America. Specimens SMNS 12478, GPIT-RE-3182 and GPIT-RE-3184, belonging to *Meyerasaurus*, *Peloneustes* and *Liopleurodon*, respectively, were examined first-hand and measurements and photographs were taken to inform the modelling. The digital models of *Dolichorhynchops*[55,56], *Thalassomedon*[54,57] and *Hydrotherosaurus*[54] were based on measurements,

photographs and two-dimensional reconstructions from the literature (Supplementary Methods). The digital models of three derived ichthyosaurs, previously published in Gutarra et al.[29], are based on almost complete specimens: *Temnodontosaurus platyodon* (NHMUK 2003), a neoichthyosaurian from the Early Jurassic of the U.K.; and two thunnosaurian ichthyosaurs, *Stenopterygius quadriscissus* (NHMUK R4086) and *Ophthalmosaurus icenicus* (NHMUK PV R3702, R3898, R4124) from the Early Jurassic of Germany and the Middle–Late Jurassic of the UK, respectively. Additionally, three extant cetaceans were included in this study, the odontocetes *Tursiops truncatus* (model previously described in Gutarra et al.[29]) and *Orcinus orca*, and the mysticete *Megaptera novaeangliae*. Life reconstructions in lateral and dorsal views, as well as photographs from lives specimens, provided the information to model the body and appendages of the bottlenose dolphin *Tursiops truncatus*[58] and the killer whale *Orcinus orca*[58,59]. The body and appendages of the humpback whale *Megaptera novaeangliae* were digitally modelled using an aerial photograph of the dorsal aspect of a wild specimen[60], as well as published information on the planform and cross-sectional shape of the flippers[61]. Digital models for all taxa were created as NURBS geometries using Rhinoceros v. 5 (Supplementary Methods, Supplementary Fig. 8) and are available for download at https://doi.org/10.5281/zenodo.5979631. The geometric parameters of the models (i.e. surface area and volume) were calculated using Rhinoceros measuring tools.

### Computational fluid dynamics. 
Computer flow simulations were carried out using the commercial software ANSYS-Fluent (v. 18.1 Academic). Our protocol has been validated (i.e. computed drag compared to data from water tank experiments) for external flow over slender bodies[29], and proved to replicate with an accuracy ≥ 95% the drag of standard rotational bodies of varying fineness ratios, within a broad range of Reynolds numbers (Re)[29]. We used the fully-turbulent shear stress transport (SST) model to solve the Reynolds-averaged Navier–Stokes (RANS) equations, as the Reynolds numbers of our analyses fall within the turbulent flow regime (> $10^6$).

Models were imported into ANSYS, where a cylindrical enclosure and refinement box around the wake area were created using the geometry tools. As all taxa are bilaterally symmetrical, only half of the models and the enclosing domain were used in the simulations to economise on computational resources. The virtual flow domain was then meshed with the ANSYS meshing tool, producing grids of 5–15 million elements, depending on the geometry, that combined tetrahedral elements in the region of free flow, and 20–25 layers of prismatic elements in the boundary layer region (i.e. the area adjacent to the non-slip wall surface). Flow was simulated using a double precision, stationary pressure-based solver and a second-order discretization method. Convergence (i.e. the point where the simulation reaches a stable solution) and mesh independence (the influence of mesh size on results) were tested. The total drag coefficient ($C_d$), as well as the coefficients for its internal components, the viscous drag ($C_f$) and the pressure drag ($C_p$) were calculated using the formula:

$$C_x = 2D / \rho\, u^2 S$$

where $D$ is the drag force in N (total, viscous or pressure drag respectively), $\rho$ is the density of water, 998.2 kg m$^{-3}$ at 20 °C; $u$ is the inlet velocity in ms$^{-1}$ and $S$ is the wetted surface area of the model in m$^2$ (see Supplementary Table 2 for a sensitivity test on the flow physical parameters). Because the drag coefficient decreases with increasing Re, the comparison of $C_d$ between taxa was done in conditions of dynamic similarity (i.e. same Re, that is same length and same velocity). The Reynolds numbers used here, $5 \times 10^6$–$10^7$, covers the range of Re values at which our selected animals likely moved, from about $2 \times 10^6$ for *Stenopterygius*, to $1.6 \times 10^7$ for *Megaptera*, considering a conservative velocity of 1 ms$^{-1}$. Additionally, the $C_d$ for $5 \times 10^6$–$10^7$ is numerically close to the average $C_d$ of the much wider range $10^6$–$5 \times 10^7$ used in a previous study[29]. The drag results presented here correspond to conditions of zero lift, to eliminate potential variability in the results caused by induced drag. When required, small adjustments were made to the orientation of the models relative to the incoming flow so that the lift remained close to zero.

Our protocol using 3D static CFD simulations provides an objective assessment of the influence of morphology and size on drag forces independent of motion, as well as allowing for comparisons of wide arrays of taxa[29]. Moreover, aquatic animals commonly use inertial displacement in the absence of movement (i.e. gliding) during submerged swimming to economise energy[62–64]. Computed drag coefficients of dolphins obtained with this method are consistent with estimates obtained from gliding dolphins in water tanks[33].

### Drag-associated energy costs of steady swimming (drag per unit of volume). 
The drag per unit volume represents the contribution of drag to the cost of locomotion (i.e. the energy spent transporting a unit of mass a unit of distance[37]) in steady swimming (i.e. constant speed, when thrust equals drag force), also called here $COT_{drag}$. The total metabolic cost of transport results from dividing the total power ($P_{in}$) by the mass ($m$) and the velocity ($u$):

$$COT = P_{in} / m\, u$$

and therefore, $COT_{drag}$ can be obtained by dividing the drag power ($P_{drag}$) by the mass and the velocity,

$$COT_{drag} = P_{drag} / m\, u = D / \rho V$$

where the volume ($V$), can be considered a proxy for body mass (assuming similar body density). For pelagic swimming animals it is reasonable to assume an approximate body density close to that of sea water[65], consistent with measurements from living cetaceans[66,67] and estimates of density in extinct marine reptiles[57]. Note that our calculations consider only the mechanical expenses of locomotion and do not account for the power invested in maintaining the basal metabolism or losses due to muscle efficiency[43].

The drag per unit of volume ($COT_{drag}$) was calculated for an inflow velocity of $1\ ms^{-1}$, for models scaled to equal total volume, in order to estimate potential differences in drag-associated costs from body shape alone. Simulations excluding the limbs were added to evaluate the contribution to drag of limbs and body separately. To test the effect of body size on $COT_{drag}$, simulations on full models at life-size scale were carried out for an inflow velocity of $1\ ms^{-1}$. The total body length in the fossil taxa is the average of all available specimens for each genus, with values obtained from personal observations or from the literature (Supplementary Data). The range of sizes for adults of living cetaceans were taken from the literature (Supplementary Data). This study is not concerned with absolute values of the drag power, as it is known that the drag estimated from rigid bodies is smaller than dynamic drag. Hence results are normalised to the values obtained for the bottlenose dolphin *Tursiops*, here used as a reference.

Comparisons presented herein do not account for propulsive efficiency ($\eta$). The reason for this is that there are no clear differences in this parameter between highly specialised caudal oscillation and underwater flying. Estimates from large extant aquatic tetrapods displaying these two swimming styles, such as cetaceans and sea lions, have produced similarly high values of $\eta$, 0.8 and 0.75–0.9 respectively[16]. Potential differences in performance due to the kinematics and the shape of propulsive elements should not be dismissed, however, they cannot be included in this model based on current knowledge.

**Effect of neck anatomy on the drag of plesiosaurs**. Flow simulations were performed for a set of plesiosaur models with varying neck lengths at a constant $Re = 5 \times 10^6$ to compute the total drag coefficient ($C_d$) as well as the coefficients for skin friction ($C_f$) and pressure drag ($C_p$). These models were built in Rhinoceros v. 5.0 by enlarging the neck of a basal plesiosaur, the rhomaleosaurid *Meyerasaurus victor* (Supplementary Fig. 8d), in which the ratio of neck length-to-trunk length (hereafter neck ratio) is 0.87, to encompass neck ratios of 2×, 3×, 4×, 5× and 7×. *Meyerasaurus* was chosen because of its plesiomorphic characteristics among plesiosaurs and relatively short neck. As shown in the Results, the drag of the limbless bodies does not differ significantly between plesiosaur models (Fig. 1a, Fig. 2a), thus making this model representative of a general plesiosaur morphology. We measured the neck as the distance from the base of the head to the edge of the pectoral girdle and the trunk as the distance between the acetabulum and the glenoid (i.e., inter-girdle distance). Elasmosaurinae, a subfamily of elasmosaurs[52], are the plesiosaurs with the longest necks described so far, with neck ratios from 2 to 3.7[23] (Supplementary Data). Therefore, neck ratios up to 4× correspond to proportions observed in nature, while neck ratios above 4× represent hypothetical body shapes. Previous work suggested that thicker neck contours provide a hydrodynamic advantage in plesiosaurs by reducing the drag coefficient[20]. To control for the impact of neck thickness on our results, we performed sensitivity tests accounting for this parameter (Supplementary Fig. 9).

Next, the drag at a constant inflow velocity of $1\ ms^{-1}$ was estimated for the same models of variable neck proportions, this time scaled to a constant trunk length. The total computed drag was then divided by the volume of the trunk. This analysis was aimed to test whether the enlargement of the neck length while maintaining a constant size of trunk would at some point become energetically costly. The trunk contains the muscles involved in locomotion and has previously been considered a better proxy for total body size than total length[46] because of the enormous variation in body proportions in sauropterygians.

Finally, to test the interplay of neck ratio and body size a final set of simulations was carried out for various models of plesiosaurs at life-size dimensions, including seven extra limbless plesiosaur models to ensure a better representation of trunk sizes and neck proportions in derived plesiosaurs. We used photographs or reconstructions from the literature and where possible, material obtained from personal observation, of well-preserved adult specimens of the basal plesiosaurs *Rhomaleosaurus thorntoni* (reconstruction by Smith & Benson[68]) and *Plesiosaurus* sp. (3D mounted skeleton in the National Museum of Wales, personal observation); the pliosauroid *Kronosaurus boyacensis* (MJACM1[69]); the polycotylid *Polycotylus latippinus* (reconstruction by O'Keefe and Chiappe[70]); the elasmosaurine elasmosaurs *Styxosaurus* sp. (SDSM 451[71]) and *Albertonectes vanderveldei*[44]; and finally the short-necked aristonectine elasmosaur *Aristonectes quiriquinensis* (SGO.PV.957[45]).

The drag force was computed from simulations at the same inlet velocity of $1\ ms^{-1}$, then divided by the volume of the trunk for each model and finally normalised to the results of the *Meyerasaurus* model. The relative values of drag-per-unit-trunk-volume ($D/V_{tr}$) were visualised with a colour gradient over a two-dimensional plot of neck ratio and trunk length. Contour lines showing interpolated values for the rest of the plot area were added using the function geom_contour of the package ggplot2 v.3.3.2[72] in R v.3.6.2[73]. Additionally, the drag-per-unit-trunk-volume was plotted against the trunk length, and the correlation between the log-transformed variables was calculated using Pearson's product moment correlation coefficient. A univariate linear regression model was

fitted using the R package moonBook[74]. The resulting linear equation $\log(y) = a\log(x) + b$, was finally transformed into a power equation to represent the relationship between the non-transformed variables, $y = e^b + x^n$.

**Evolutionary analysis of neck proportions and trunk length in sauropterygians**. We compiled a dataset of neck and trunk lengths for 79 sauropterygian species through the Mesozoic (Supplementary Data). Neck ratios and trunk lengths were visualised in a bivariate morphospace showing areas occupied by different sauropterygian clades. Functional explanations of the two axes on drag-per-unit-volume are based on previous CFD analyses on real and hypothetical models. The trunk length (cm) and neck proportions were plotted on a phylogeny of sauropterygians and values were inferred at ancestral nodes using the R package phytools v0.7-47[75]. For this, an informal composite sauropterygian tree was assembled based on a published phylogeny[6] that combines phylogenetic relationships of Plesiosauria[76] and Triassic sauropterygians[77] (Supplementary Methods).

Rates of evolution were estimated for trunk length (using raw and $\log_{10}$-transformed data) and neck ratio on 40 time-calibrated phylogenies in a Bayesian framework, using the variable-rates model of BayesTraits v.2.0.2[78] and R v.3.6.2, with code from Stubbs et al.[79]. The sauropterygian tree was time-calibrated 20 times, using both the cal3[80,81] and Hedman[82] scaling methods to account for uncertainties of fossil occurrences and differences in dating models (Supplementary Methods). Evolutionary rate heterogeneity was evaluated for the 20 trees from each dating approach with a reversible jump Markov Chain Monte Carlo algorithm (rjMCMC) using default prior distributions. The analysis used 2 billion iterations, of which the first 400 million were discarded as burn-in, and parameters were sampled every 80,000 iterations. The method detects shifts in rates of evolution by incorporating branch-specific scalars and rescales branch lengths that deviate from expectations of a homogeneous Brownian motion (BM) model. Rates results were summarised using consensus trees derived from the 20 iterations of each dating approach using phytools[75], in which colours of branches represent mean rate scalars from all 20 trees. Convergence was tested using the minimum effective sample size function of the R package CODA v.0.19.3[83]. The fit of a heterogeneous variable-rates model was tested against a null homogeneous random walk model (BM) using log Bayes factors[48] (BF), calculated from the marginal likelihoods of these two models, obtained using the stepping-stone sampling method, with 100 stones per run for 1000 iterations[84]. Lastly, a randomisation test was applied to test for significantly different rates of evolution in Elasmosauridae, Polycotylidae and Thalassophonea. This analysis performs multiple random samplings of mean rate values and calculates differences between sampled and then non-sampled branches over 9999 replicates[85].

**Reporting summary**. Further information on research design is available in the Nature Research Reporting Summary linked to this article.

## Data availability
Supplementary Information includes Supplementary Figures, Supplementary Tables and Supplementary Methods; Supplementary Data includes calculations supporting the present results and datasets used in the evolutionary rates analyses. These files, as well as the digital models created for the computer flow simulations performed in this study, can be found in the GitHub repository (https://github.com/SusanaGutarra/Plesiosaur-hydrodynamics-evolution) and at https://doi.org/10.5281/zenodo.5979631[86].

## Code availability
The authors declare that the code supporting the findings of this study is available at https://doi.org/10.5281/zenodo.5979631[86].

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

## Acknowledgements

We thank Imran Rahman (Natural History Museum, London), Stephan Lautenschlager (University of Birmingham) and Emily Rayfield (University of Bristol) for comments and discussions on early versions of this manuscript. We are grateful to Armin Elsler (University of Bristol) for sharing custom R-code to extract and visualise the results of the BayesTraits evolutionary rates analysis. We thank Ingmar Werneburg (Palaeontology Collection, Tübingen, Germany), Erin Maxwell (Staatliches Museum für Naturkunde Stuttgart, Germany) and Caroline Buttler (National Museum of Wales, UK) for facilitating access to fossil marine reptile specimens. We would also like to thank Neil Kelly, and an anonymous reviewer for constructive comments that helped improve this manuscript. S.G. was supported by the Natural Environment Research Council and the GW4+DTP (grant no. NE/L002434/1), with additional support from CASE partner National Museum of Wales; B.C.M. and T.L.S. are funded by NERC grant NE/P013724/1 and M.J.B by ERC grant 788203 (INNOVATION).

## Author contributions

S.G. designed and executed the CFD simulations, compiled the phenotypic data, assembled the sauropterygian tree and wrote the main manuscript. C.P. helped design and supervised simulation analyses; M.J.B, B.C.M. and T.L.S. helped design and supervised the evolutionary rates analyses. T.L.S performed the dating and rates analyses. All authors contributed to the final manuscript.

## Competing interests

The authors declare no competing interests.
