## [Peer Review File · Communications Biology]

Reviewers' comments:

Reviewer #1 (Remarks to the Author):

The manuscript details a very nice study on the comparative morphology of extinct aquatic reptiles and modern cetaceans. The paper is really divided into two parts. The first part is an analysis of the hydrodynamics in relation to morphological variation among the various species. The second part is an examination of the evolutionary rates of the relative length of the neck to trunk length of plesiosaurs. I cannot speak critically on this latter examination, but it seems to be fully justified. I can speak to the hydrodynamic morphology of the various secondarily aquatic marine vertebrates. For the first time, the authors have tried to get a handle on the potential liability of a long neck. The CFD appears to give rigorous results for the assumptions that were used. The results are interesting but there are a number of minor points including corrections and clarifications that need to be addressed and one potentially critical point that also needs to be addressed. This critical point is in regard to the scale that was varied to make comparisons between the species. Initially as standard hydrodynamic analysis would dictate, the drag coefficient (C_d) was calculated from CFD after 3D reconstruction of the animals and standardizing the swimming velocity at 1 m/s. The analysis was later scaled by using equivalent volumes for the animals. First, the velocity would be considered very low as the routine swimming speeds of cruising cetaceans would be around 2-3 m/s. The low swimming speed would be appropriate for plesiosaurs and ichthyosaurs, which were computed by Massare and Motani. This may make a difference regarding the hydrodynamics and the optimization of the body morphology. It is the use of equivalent volumes for scaling that may not provide the desired results of differences in body morphology and the effects on hydrodynamics. For a relative assessment of C_d , it would probably have been better to use equivalent surface areas as drag is equal to the product of $0.5 \rho v^2 C_d A$, water density, C_d , velocity squared and wetted surface area. As the velocity and density are already standardized, an equivalent surface area would give a C_d based on only differences in body geometry (not considering surface texture, which is largely unknown). Standardizing the surface area will provide a better evaluation of the differences in geometry among the species. Frictional drag is based on the skin friction of the wetted surface area, whereas the pressure component of drag is dependent on body contours. In consideration of C_d and fossil species the authors missed an important reference, where scale models of ichthyosaurs and a plesiosaur were empirically tested in a water tunnel and could be used to help validate the CFD analysis. The reference is: Aleyev, Y. G. (1977). Nekton. The Hague, Junk.

and the C_d values are 0.005 for *Stenopterygius*, 0.004 for *Eurhinosaurus*, and 0.009 for *Cryptoclidus*. In estimates of C_d in this study, some clarification is needed. In the Methods, it is stated that a 3D geometry for the various species was derived. Was the 3D geometry used in calculation with CFD? How was the wetted surface area measured from the 3D geometry? All the figures displayed only lateral views of the animals. This is particularly important in Fig. 1, which shows the flow regime. A comparative figure showing the flow around the dorsal view must be provided. None of these animals is truly axisymmetrical. Indeed in Fig. 4, the illustrations appear to show an axisymmetrical body shape for all the species. If this is not so, then it should be indicated in the figure legend whether the reader is looking at a lateral or dorsal view. Another potential source of error is the density of water and the animals. In the model, the density of water was taken as 998.2 kg m⁻³ (li. 416), which is the approximate density of freshwater, while all the species were marine (1025 kg m⁻³). There was also the assumption that the density of the reptiles was equivalent to the cetaceans, which may not be the case. Although the analysis is mainly good, the paper needs to acknowledge potential sources of error, clarify measurements and assumption, and standardize surface area to assess the effect of body form.

Minor points;

127-128- In general, caudal oscillatory swimmers are faster than pectoral oscillators. This is generally known and should be acknowledged and referenced.

Fig. 2A- The relative drag per unit volume is not a cost of transport. Cost of transport is the energy to move a unit mass, a unit distance. Remove "(COTdrag)" from the Y-label.

252- replace "huge" with "added"

255-257- This is an inaccurate statement. The only information on the velocity when turning by sea lions was the maximum velocity. Nothing was provided regarding reduction of speed and flow

separation.

257-258- Yes, more information is needed on the role of the neck. Did the authors consider that the neck of long-necked plesiosaurs may have used the neck to snake through the water, which could either added to the drag or improve thrust production? What information can be found on the flexibility of the neck?

432- Change 'porpoise' to "dolphin". The animal tested by Lang and Daybell was a Pacific white-sided dolphin (*Lagenorhynchus obliquidens*), not a porpoise which is in another taxonomic family. Although Lang and Daybell referred to the animal as a porpoise, the misnomer should be corrected and not continued.

437- change to "The total metabolic cost of transport results..."

460- Change to "bottlenose"

Reference 23- What is the full reference?

Additional references to be considered in Discussion include:

Gutarra, S. and Rahman, I. A. 2021. The locomotion of extinct secondarily aquatic tetrapods. *Biological Reviews* doi: 10.1111/brv.12790.

Gough, W. T., Smith, H., Savoca, M., Czapanskiy, M., Fish, F., Potvin, J., Bierlich, K. C., Cade, D., Di Clemente, Kennedy, J., Segre, P., Stanworth, A., Weir, C. and Goldbogen, J. A. 2021. Scaling of oscillatory kinematics and Froude efficiency in baleen whales. *Journal of Experimental Biology* 224: jeb237586.

Reviewer #2 (Remarks to the Author):

Gutarra Díaz and colleagues used 3D models of plesiosaurs, ichthyosaurs and cetaceans to compare how body shape and proportions influence drag within simulated flow experiments (computational fluid dynamics or CFD). They also looked at evolutionary rates of body proportions (trunk size and neck to trunk ratio) among sauropterygians.

Their key findings are that although plesiosaurs do experience higher drag than ichthyosaurs or cetaceans due to their unique morphology, these differences are relatively minor. Furthermore, when size is taken into account, differences between smaller and larger members of a given clade are more significant than differences among clades/morphotypes. They use these results to infer that plesiosaurs were not necessarily significantly slower swimmers than ichthyosaurs and cetaceans in contrast with some previous suggestions. They also find little correlation between fineness ratio (ratio of body length to diameter) and drag, suggesting that this ratio, which has been widely used to estimate swimming performance across marine vertebrates, may not be a useful indicator of swimming performance after all.

Finally, a comparison of evolutionary rates of trunk size and neck length in sauropterygians produces somewhat complex results but indicates an increase in neck length evolution among elasmosaurines and they further speculate that large body size in elasmosaurines helped to increase overall locomotory efficiency (as estimated by drag per unit volume) which could have offset increased drag induced by longer necks, but only up to a certain ratio which might suggest a limit to how long plesiosaur necks could get.

Overall this is a strong paper with clearly expressed and tested hypotheses and a solid design. There are inherent limits to what can be inferred from testing drag on static models, but it produces useful and quantitative comparisons. This is far from the final word on plesiosaur swimming, a topic that has been debated and investigated for many years, but it is a significant contribution to our understanding. I recommend acceptance for publication with a handful of relatively minor comments detailed below.

Detailed comments:

Line 27 - 'important' is a bit subjective, I would suggest being more specific in how you are assessing importance here: diversity? clade longevity?

Line 29 - 'medium to large body sizes' I suggest specifying an approximate size range here if possible.

Line 30 - missing 'this' after achieved?

Line 35-37 - although enigmatic, the locomotory mode of plesiosaurs has been the subject of extensive research and it might make sense to cite a few of those papers here (in parallel with the ichthyosaur citations), although I expect they are cited later in the text.

Line 71 - "neck plasticity" here is a bit confusing of wording, I think you mean it in an evolutionary sense across the clade (i.e. the wide range of neck to trunk proportions observed across sauropterygia), but plasticity has other meanings based on material properties as well as ecology, development or phenotype within an individual or across a population or taxon. Perhaps you can reword a bit for clarity? In particular, the flexibility of plesiosaur necks has been the subject of study, I don't think that's what you are talking about here but I'm not certain.

Line 85 - I guess this is true for all models (cetaceans, ichthyosaurs & plesiosaurs)? It might not hurt to state that explicitly in this sentence.

Something of an aside, from Fig 1b it looks like the anterior stagnation point for cetaceans is notably more pronounced than in the reptiles. I suppose because they are more blunt headed, possibly related to higher mammalian Encephalization Quotients. Maybe not relevant to the focal questions here but an interesting result.

Line 123 - 124 - Another aside here, but the notable difference between ichthyosaurs and cetaceans is that parvipelvians reduce the hind limbs but not to the extent that cetaceans do. I'm curious if you tested the impact of removing just the hind limbs of the ichthyosaurs and if it made any notable difference in drag?

Likewise, there is a degree of uncertainty regarding the upper lobe and dorsal fin proportions in most ichthyosaurs, apart from those taxa with very good soft tissue preservation, so I am curious how big of an impact the assumed form of these appendages have on drag calculations? Possibly you addressed this in your earlier work on ichthyosaurs, or in the supplement which I have not gotten to yet, so apologies if these are questions you have already tackled elsewhere.

Line 126 - more succinct/direct phrasing would be "morphology of plesiosaurs produced higher drag than parvipelvic ichthyosaurs..." rather than the quasi double-negative here.

Line 145-6 (OK, this partly answers my questions from line 123 above!)

Line 221 - higher drag than other plesiosaurs of the same size right? It might be useful to specify.

Line 253 - I buy the argument here BUT as the authors already pointed out in their introduction, comparison with living forms is hampered by the fact that there are no good extant analogues for plesiosauromorphs. I would perhaps suggest replacing 'This is improbable because' (which implies a degree of quantitative hypothesis testing) with 'However,'

Line 258 - 260 - Again I agree with the overall sentiment here, but I worry that the case is overstated just a bit. The authors have not done direct testing of plesiosaur swimming performance w.r.t thrust generation, rather they compared the drag produced by static models. Thus I don't think there is strong enough evidence to 'reject' slow swimming speeds in plesiosauromorphs, rather they have shown that the drag produced by the plesiosauromorph body plan would not preclude them from faster swimming speeds.

So changing "rejected" to "not supported" would probably be more appropriate.

Line 303 - This pattern of increasing body size over time is typical for marine tetrapods and has been investigated extensively (e.g. Gearty et al. 2018 for a recent paper on marine mammals). It would probably be worth discussing the results presented here within the context of that larger evolutionary pattern.

Line 330 - Misspelled should be "Weddellonectians"

Line 344 - Here's a case where I think you are *understating.* Hydrodynamics almost certainly places significant constraints on neck morphology in animals that swim fully submerged. However, wording is fine as is if it is to your preference.

Figure 5a - very minor suggestion: inverting the order on the key will match the order in the univariate plot which would ease interpretation somewhat

5d - The log10 plot is very difficult to read at this small scale. I suggest adjusting the elements in 5d & c so it can be enlarged (for example you don't need to reproduce the slow to fast color scale bar 3 times & it could be moved to the lower left to make more room for the log10 plot.

Alternatively it can be omitted as it is already in the supplement and does not have major impact on the key results discussed here.

Line 384 - 'novaengliae' should be 'novaeangliae'

RESPONSE TO REVIEWERS

Referee expertise:

Referee #1: marine animal locomotion, computational fluid dynamics

Referee #2: marine tetrapod evolution

Reviewers' comments:

Reviewer #1 (Remarks to the Author):

The manuscript details a very nice study on the comparative morphology of extinct aquatic reptiles and modern cetaceans. The paper is really divided into two parts. The first part is an analysis of the hydrodynamics in relation to morphological variation among the various species. The second part is an examination of the evolutionary rates of the relative length of the neck to trunk length of plesiosaurs. I cannot speak critically on this latter examination, but it seems to be fully justified. I can speak to the hydrodynamic morphology of the various secondarily aquatic marine vertebrates. For the first time, the authors have tried to get a handle on the potential liability of a long neck. The CFD appears to give rigorous results for the assumptions that were used. The results are interesting but there are a number of minor points including corrections and clarifications that need to be addressed and one potentially critical point that also needs to be addressed.

We are very thankful for the positive comments, and all suggestions put forward by the reviewer. We have carefully addressed them all, as explained in detail here below. Importantly, we provide thorough theoretical justification and additional new analyses where necessary, including various new supplementary figures and tables, to address the potential sources of uncertainty or variation. We believe these have contributed to a greater clarity in the methodological background and the interpretation of our results.

This critical point is in regard to the scale that was varied to make comparisons between the species. Initially as standard hydrodynamic analysis would dictate, the drag coefficient (C_d) was calculated from CFD after 3D reconstruction of the animals and standardizing the swimming velocity at 1 m/s. The analysis was later scaled by using equivalent volumes for the animals.

First, the velocity would be considered very low as the routine swimming speeds of cruising cetaceans would be around 2-3 m/s. The low swimming speed would be appropriate for plesiosaurs and ichthyosaurs, which were computed by Massare and Motani. This may make a difference regarding the hydrodynamics and the optimization of the body morphology.

We agree with the reviewer in that some large animals might have routine (or optimal) cruising speeds that are higher than 1m/s. However, our chosen velocity is only a reference speed and was not intended to represent the optimal speed, as we explicitly state in the text (line 205). Given that preferred speeds in aquatic vertebrates display a narrow variation for a very wide range of sizes, we took 1m/s because it's a speed we can reasonably assume all the animals in our analysis could perform. We understand that this perhaps was not entirely clear, so we modified the text to clarify this point (see lines 205, 208).

Additionally, the concern expressed by the reviewer that using a different velocity could impact the results has been addressed by a new supplementary figure (Supplementary Fig. 4, lines 209–211). A higher velocity (i.e. 2 m/s) only increases the absolute drag, leaving the relative drag/volume unchanged. The raw data for this supplementary graph can be found in the Supplementary data, as well as the per-group comparison showing no changes in the results.

It is the use of equivalent volumes for scaling that may not provide the desired results of differences in body morphology and the effects on hydrodynamics. For a relative assessment of C_d , it would probably have been better to use equivalent surface areas as drag is equal to the product of 0.5, water density, C_d , velocity squared and wetted surface area. As the velocity and density are already standardized, an equivalent surface area would give a C_d based on only differences in body geometry (not considering surface texture, which is largely unknown). Standardizing the surface area will provide a better evaluation of the differences in geometry among the species. Frictional drag is based on the skin friction of the wetted surface area, whereas the pressure component of drag is dependent on body contours.

We agree with the reviewer that the matter of scaling is one of great importance and needs to be carefully considered to avoid confounding effects. First, here below we clarify our scaling choices in the two metrics we use:

1. Drag coefficient (C_d): a non-dimensional measure of the drag for a given Reynolds number. By definition ($C_d = \frac{2D}{\rho u^2 S}$) the drag coefficient is non-dimensionalised relative to a reference surface. The surface of choice, depending on the focus of the study, can be frontal area, surface area, and in other instances volume^{2/3}. In our case we use total surface area (i.e. wetted surface), as is standard practice when assessing external flow over slender geometries. This means that our comparisons of C_d (Fig.1) already include a standardisation to surface area.
2. Drag per unit of volume: this is calculated for a reference speed of 1m/s and presented as relative values (Fig. 2). In the methods section, we derive how drag divided by volume is a proxy for the contribution of drag to the cost of transport, a mass-normalised metric for energetic performance, and therefore establish its biological relevance.

We understand from the reviewer's comment, that they suggest we compare this second metric between animals scaled at the same surface area. Here below we explain in detail why the use of volume as a scaling criterion is entirely justified and provides the best basis for assessing shape and size effects in the energy expenses of steady swimming:

- We used volume scaling in a previous publication by our group (Gutarra *et al.*, 2019), and there we discussed in length its advantage over scaling to body length, another scaling parameter sometimes used in biomechanical studies.
- Surface scaling would provide a very similar trend to that obtained by volume scaling. The explanation to this can be found in our own results: we show that because the drag is mainly frictional in these slender forms (see Supplementary Fig 2), the differences we observe between taxa highly influenced by the ratio of surface-area-to-volume (Fig. 2f). This is the basis of the differences in drag/volume between animals of different sizes: as animals get larger, volume scales to the power of 3, while surface scales to the power of 2. If we were to scale to equal surface instead of equal volume, we would just capture another angle of this phenomenon, and would not add anything new to the current analysis. We illustrate this point with a graph comparing the two alternative scaling criteria (volume versus surface) for the analysis in Figure 2a (see below plot). The per-group differences are slightly enhanced when using the surface-scaling, but the general trend remains.

- Finally, we'd also like to argue that standardising to volume (a proxy for body mass) is a more straightforward concept for the reader, as it answers the question "what morphology distributes a given mass in a more hydrodynamically efficient way?"

We believe the explanations provided above fully justify that there is no need to replace the scaling criterion in our analyses.

In consideration of Cd and fossil species the authors missed an important reference, where scale models of ichthyosaurs and a plesiosaur were empirically tested in a water tunnel and could be used to help validate the CFD analysis. The reference is:

Aleyev, Y. G. (1977). Nekton. The Hague, Junk. and the Cd values are 0.005 for Stenopterygius, 0.004 for Eurhinosaurus, and 0.009 for Cryptoclidus.

We thank the reviewer for suggesting additional reference on experimental results from model animals. However, after carefully assessing the data, we concluded that the values of drag coefficient presented in this reference cannot be directly compared with ours. Here below, we elaborate on this:

The values presented by Aleyev, 1977 for multiple taxa (table 17, page 243) represent only the pressure drag component of the drag coefficient, C_{Dp} (C_p in our manuscript, see methods section, line 442) and not the total drag coefficient (C_d). When we compare these estimates to our computed pressure drag coefficients, we noted that Aleyev's numbers are unusually high. For example, taking *Stenopterygius*, the only taxon we have in common: Aleyev estimates $C_p=0.005$ (for $Re= 1.2 \times 10^7$) and our computed value is $C_p=0.00126$ (for $Re= 10^7$).

Our CFD method has been validated against experimental data from water tank experiments (Gutarra *et al.*, 2019) and it can simulate accurately the proportion of pressure and skin friction drag in torpedo-like forms (see also Supplementary Fig. 2a, comparing our computed skin friction drag coefficients (C_f) to the estimates provided by the ITTC57 empirical formula).

Alayev (1977) also used an empirical formula, most likely the Prandtl formula, to estimate C_f , and then subtracted it from the total drag coefficient (C_d) to obtain the pressure drag coefficient (C_p). For *Stenopterygius*, $C_f = 0.0029$ (based on ITTC57) or 0.0027 (based on Prandtl's formula) at $Re = 1.2 \times 10^7$. This means their pressure drag component is 64% of the total drag, much higher than expected from a fully submerged slender morphology, for which the skin friction component should be the majority of drag (Vogel, 1994; Hoerner, 1965; Schlichting, 2017).

There are various reasons why Aleyev's experimental data overestimates the pressure component: a) their models might not be submerged at enough depth which adds extra wave drag (their *Stenopterygius* model measures 2.1 m and is only submerged at a depth of 0.6 m); b) it is not clear from the text if the drag incurred by the supporting strut has been subtracted; c) they assume the induced drag (a by-product of lift) is zero, but often physical models need small adjustments to the angle of attack so that lift is zero.

For all reasons mentioned above, we would rather not cite this reference. However, we have included other references. At the beginning of the Results section, we show there is good agreement between the computed drag coefficient of our dolphin model ($C_d = 0.00413$ for $Re = 10^7$) with other estimates from the literature: the drag coefficient of a gliding dolphin ($C_d = 0.0034$ for $Re = 9.1 \times 10^6$, by Lang & Daybell, 1963) and the drag coefficient from other static CFD analyses ($C_d = 0.004$ $Re = 10^7$, by Riedeberger & Rist, 2012) (see lines 86–91).

In estimates of C_d in this study, some clarification is needed. In the Methods, it is stated that a 3D geometry for the various species was derived. Was the 3D geometry used in calculation with CFD? How was the wetted surface area measured from the 3D geometry?

Indeed, we use the 3D geometries generated in Rhinoceros to perform the CFD simulations. The wetted area, as well as the volume of the models were calculated using the Rhinoceros measuring tools as stated in the Table S1 description. However, to make this clearer we have inserted a phrase in the methods section as well (lines 420–421).

All the figures displayed only lateral views of the animals. This is particularly important in Fig. 1, which shows the flow regime. A comparative figure showing the flow around the dorsal view must be provided.

A supplementary figure showing an oblique dorsal view has been included in the Supplementary Information (see line 117 and new Supplementary Fig. S1).

None of these animals is truly axisymmetrical. Indeed in Fig. 4, the illustrations appear to show an axisymmetrical body shape for all the species. If this is not so, then it should be indicated in the figure legend whether the reader is looking at a lateral or dorsal view.

Our models are indeed not axisymmetric; in the figures we only display representative views of one aspect. We have modified the figure legend in Fig. 4c to clarify that models are displayed in dorsal view (see line 296–297). Additionally, we would like to highlight the fact that a link is provided in the manuscript where all 3D models can be downloaded, inspected, and used by the readers (lines 588 and 591).

Another potential source of error is the density of water and the animals. In the model, the density of water was taken as 998.2 kg m^{-3} (li. 416), which is the approximate density of freshwater, while all the species were marine (1025 kg m^{-3}).

There was also the assumption that the density of the reptiles was equivalent to the cetaceans, which may not be the case.

Density of water in the simulations: The simulations have been performed using the standard material properties for water provided by the software by default, which is water at 20°C . We understand it would be more straightforward to use seawater conditions, but here below we fully justify why this should not be a cause of concern and describe a new sensitivity test included in the

revised manuscript (Supplementary Table 2, described in Supplementary Information, Sensitivity test, page 18; see also main text, line 448):

1. On the one hand, results of drag coefficient (C_d) are independent of the density of the media used in the simulations, because they are presented in terms of non-dimensional parameters (drag coefficient C_d for a given Reynolds numbers, Re).

The Reynolds number is defined as $Re = \rho ul/\mu$. The similarity rule states that two geometrically identical objects moving at the same Re produce the same flow patterns and forces. This means that any combination of conditions that result in a given Re produce the same drag coefficient. In the new Supplementary Table 2 (also attached below) we use as example a *Liopleurodon* and *Tursiops* at life-size scale. For equal Re , the drag coefficient is the same regardless simulation is done in fresh or seawater. This is the similarity rule at work: animals moving at 1 ms^{-1} in freshwater produce exactly the same drag as when moving at 1.03 ms^{-1} in sea water.

Thanks to the similarity principle, engineers can model the drag coefficients of airplanes or ships by using small replicas in either water tanks or wind tunnels (therefore using different density media).

2. Our results of relative drag/volume would not be affected by using different water parameters. This is also illustrated in the new Supplementary Table 2. The difference in density between sea and freshwater would have a minor effect on the absolute values of drag but importantly, would have no impact in the relative ranking, which is the focus of our analysis.

Supplementary Table 2 shows that the 2.5% difference in density and 8% difference in dynamic viscosity between fresh and seawater have a trivial effect on Re for a same given velocity, which results in a small increase of the drag in absolute terms in both taxa, but importantly, do not affect the relative drag/volume.

This would stand even if differences in the density of the media were larger. For example, testing these models in an air tunnel instead of a water tank, would provide the same ranking of drag/volume.

	Comparison 1		Comparison 2		Comparison 3	
	Liopleurodon	Tursiops	Liopleurodon	Tursiops	Liopleurodon	Tursiops
	Freshwater	Freshwater	Sea water	Sea water	Sea water	Sea water
fluid density (ρ), Kg/m^3	998.2	998.2	1025	1025	1025	1025
fluid dynamic viscosity (μ), Ns/m^2	0.001002	0.001002	0.00109	0.00109	0.00109	0.00109
flow velocity (u), ms^{-1}	1	1	1.029	1.029	1	1
body length (l), m	4.88	2.85	4.88	2.85	4.88	2.85
body Volume (V), m^3	1.78	0.27	1.78	0.27	1.78	0.27
Reynolds number (Re)	4.9×10^6	2.8×10^6	4.9×10^6	2.8×10^6	4.6×10^6	2.7×10^6
Drag (D), N	35.06	8.36	35.06	8.36	31.03	7.37
Drag coefficient (C_d)	0.005355	0.005074	0.005355	0.005074	0.005399	0.005018
Drag/Volume (D/V), N/m^3	19.66	31.13	19.66	31.13	17.40	27.44
Relative Drag/Volume	0.63	1	0.63	1	0.63	1

Density of the Mesozoic marine reptiles: For swimming animals, and particularly for pelagic swimmers, using the density of water as an approximation to body density has been a widely applied assumption in biomechanics (Vogel, 1994). Examples of body density estimates from the literature vary within narrow ranges close to the density of sea water, from 1025.2 to 1043 kg/m^3 (Narazaki *et*

al., 2018; Miller *et al.*, 2016) in cetaceans, and 1015 to 1060 kg/m³ in elephant seals (Aoki *et al.*, 2011).

It is reasonable to assume that ichthyosaurs and plesiosaurs had similar body density to living pelagic swimmers. This assumption is often adopted in biomechanical research dealing with Mesozoic marine reptiles (e.g. Massare, 1988; Motani, 2001) and it is supported by a study that calculated the body density of Mesozoic marine reptiles (Henderson, 2006, table 4). In this paper, density is calculated from 3D reconstructions, assigning specific tissue densities to different body regions. For three plesiosaur taxa, *Liopleurodon*, *Cryptoclidus* and *Thalassomedon*, the total body density varies between 1028 to 1033 kg/m³ for a condition of neutral buoyancy, thus very close to the above-mentioned estimates from modern pelagic swimmers.

In conclusion, there is in fact variation in body density in living cetaceans and uncertainty in the density of the extinct taxa that is not accounted for here. However, the narrow range of this variation means that we can confidently use volume as a good proxy for body mass. Therefore, assuming equal densities for all living and extinct pelagic swimmers is still the simplest, most reasonable assumption we can work with. A paragraph has been added to the methods section to justify this assumption, adding relevant references (lines 479–481).

Although the analysis is mainly good, the paper needs to acknowledge potential sources of error, clarify measurements and assumption, and standardize surface area to assess the effect of body form.

We appreciate the positive remark of the reviewer. We believe we have thoroughly addressed every point raised by the reviewer regarding the potential sources of error/uncertainty (i.e. flow density, body density, choice of velocity) and provided solid theoretical and experimental justification for the use of volume as scaling criterion.

Minor points;

127-128- In general, caudal oscillatory swimmers are faster than pectoral oscillators. This is generally known and should be acknowledged and referenced.

We have added a phrase to the text acknowledging that the efficiency of caudal oscillation estimated is often higher than that of pectoral oscillation and we added the relevant literature (lines 140–143).

Fig. 2A- The relative drag per unit volume is not a cost of transport. Cost of transport is the energy to move a unit mass, a unit distance. Remove “(COTdrag)” from the Y-label.

We have removed this from the graphs and retained the explanation in the text that we use Drag per unit of volume as a proxy for COT_{drag} .

252- replace “huge” with “added”

We have replaced this expression as suggested (line 277).

255-257- This is an inaccurate statement. The only information on the velocity when turning by sea lions was the maximum velocity. Nothing was provided regarding reduction of speed and flow separation.

We agree with the reviewer that the reference doesn't specifically focus on the change in velocity during the turn or the amount of separation, however the authors provide a detailed description of the turning motion from their videos and observations and state that sea lions initiate turns bending the body and neck, curving smoothly and maintaining a streamlined appearance through the turn, which minimises drag and limits deceleration as direction changes (see Fish, 2003, page 671). We have removed the reference to flow separation and modified our statement to fit this description more closely (see line 281), which still illustrates the point that for plesiosaurs, bending the neck

during swimming might have aided turning without drastically increasing the drag. We also assert the role of necks need further biomechanical exploration.

257-258- Yes, more information is needed on the role of the neck. Did the authors consider that the neck of long-necked plesiosaurs may have used the neck to snake through the water, which could either added to the drag or improve thrust production? What information can be found on the flexibility of the neck?

Although an analysis of neck flexibility is outside the scope of this study, this is indeed a fascinating field for future research, so we have made sure that this is properly emphasized. We have modified the text to extend our suggestion to explore the effect of necks and in aspects of swimming other than manoeuvrability (see lines 282–283) and pointing to shape and flexibility as important factors to focus on. Further discussion of the implications of neck flexibility might end up being highly speculative. However, we make sure to acknowledge that the debate on the function of long necks is still ongoing and cite the pertinent literature.

432- Change ‘porpoise’ to “dolphin”. The animal tested by Lang and Daybell was a Pacific white-sided dolphin (*Lagenorhynchus obliquidens*), not a porpoise which is in another taxonomic family. Although Lang and Daybell referred to the animal as a porpoise, the misnomer should be corrected and not continued.

This has been changed as suggested (see line 464).

437- change to “The total metabolic cost of transport results...”

This has been changed as suggested (see lines 469).

460- Change to “bottlenose”

This has been changed as suggested (see lines 494).

Reference 23- What is the full reference?

Apologies for the missing information in this reference. This has been corrected (new reference 26, see line 651–653)

Additional references to be considered in Discussion include:

Gutarra, S. and Rahman, I. A. 2021. The locomotion of extinct secondarily aquatic tetrapods. *Biological Reviews* doi: 10.1111/brv.12790.

We thank the reviewer for suggesting this reference, which we have included in our manuscript (see lines 91 and 100).

Gough, W. T., Smith, H., Savoca, M., Czapanskiy, M., Fish, F., Potvin, J., Bierlich, K. C., Cade, D., Di Clemente, Kennedy, J., Segre, P., Stanworth, A., Weir, C. and Goldbogen, J. A. 2021. Scaling of oscillatory kinematics and Froude efficiency in baleen whales. *Journal of Experimental Biology* 224: jeb237586.

We thank the reviewer for suggesting this reference, which we have included in our manuscript in the comments regarding the constraints to very large sizes in swimming tetrapods (see line 348).

Reviewer #2 (Remarks to the Author):

Gutarra Díaz and colleagues used 3D models of plesiosaurs, ichthyosaurs and cetaceans to compare how body shape and proportions influence drag within simulated flow experiments (computational fluid dynamics or CFD). They also looked at evolutionary rates of body proportions (trunk size and neck to trunk ratio) among sauropterygians.

Their key findings are that although plesiosaurs do experience higher drag than ichthyosaurs or cetaceans due to their unique morphology, these differences are relatively minor. Furthermore, when size is taken into account, differences between smaller and larger members of a given clade are more significant than differences among clades/morphotypes. They use these results to infer that plesiosaurs were not necessarily significantly slower swimmers than ichthyosaurs and cetaceans in contrast with some previous suggestions. They also find little correlation between fineness ratio (ratio of body length to diameter) and drag, suggesting that this ratio, which has been widely used to estimate swimming performance across marine vertebrates, may not be a useful indicator of swimming performance after all.

Finally, a comparison of evolutionary rates of trunk size and neck length in sauropterygians produces somewhat complex results but indicates an increase in neck length evolution among elasmosaurines and they further speculate that large body size in elasmosaurines helped to increase overall locomotory efficiency (as estimated by drag per unit volume) which could have offset increased drag induced by longer necks, but only up to a certain ratio which might suggest a limit to how long plesiosaur necks could get.

Overall this is a strong paper with clearly expressed and tested hypotheses and a solid design. There are inherent limits to what can be inferred from testing drag on static models, but it produces useful and quantitative comparisons. This is far from the final word on plesiosaur swimming, a topic that has been debated and investigated for many years, but it is a significant contribution to our understanding. I recommend acceptance for publication with a handful of relatively minor comments detailed below.

We are appreciative of the positive feedback and the kind comments by the reviewer and do hope this work will contribute to expand the physics-based knowledge on the swimming of extinct marine reptiles.

Neil Kelley

Detailed comments:

Line 27 - 'important' is a bit subjective, I would suggest being more specific in how you are assessing importance here: diversity? clade longevity?

Their exceptionality is indeed related to both diversity and longevity, so this has been rephrased accordingly, and the relevant citations have been added (see line 28).

Line 29 - 'medium to large body sizes' I suggest specifying an approximate size range here if possible. We have specified a range based on total body length (2m and above) and cite a relevant previous reference (Massare, 1988) (see lines 29–30).

Line 30 - missing 'this' after achieved?

Apologies for the omission, this word has been added (see line 31).

Line 35-37 - although enigmatic, the locomotory mode of plesiosaurs has been the subject of extensive research and it might make sense to cite a few of those papers here (in parallel with the ichthyosaur citations), although I expect they are cited later in the text.

We agree with the reviewer that in this point of the text it makes sense cite the research done in the biomechanics of plesiosaur swimming. We have modified our statement to express that despite the various studies in this area published in recent years, many aspects of the locomotion of plesiosaurs remain enigmatic. A few relevant citations have been added (see lines 37–38).

Line 71 - “neck plasticity” here is a bit confusing of wording, I think you mean it in an evolutionary sense across the clade (i.e. the wide range of neck to trunk proportions observed across sauropterygia), but plasticity has other meanings based on material properties as well as ecology, development or phenotype within an individual or across a population or taxon. Perhaps you can reword a bit for clarity? In particular, the flexibility of plesiosaur necks has been the subject of study, I don’t think that’s what you are talking about here but I’m not certain.

Indeed, we use here plasticity to describe the highly variable neck proportions observed in this clade. We have added this clarification (see line 73) to avoid any potential confusion.

Line 85 - I guess this is true for all models (cetaceans, ichthyosaurs & plesiosaurs)? It might not hurt to state that explicitly in this sentence.

This is correct, all these flow characteristics can be observed in all models from the three groups. We have modified this phrase so that this is explicitly stated (see line 92).

Additionally, we refer the reader to a new supplementary figure (New Fig. S1), added in response to an observation by Reviewer 1, that shows another cross-section of the flow across a dorsal plane.

Something of an aside, from Fig 1b it looks like the anterior stagnation point for cetaceans is notably more pronounced than in the reptiles. I suppose because they are more blunt headed, possibly related to higher mammalian Encephalization Quotients. Maybe not relevant to the focal questions here but an interesting result.

That is a very valid and interesting observation. In fact, a few aspects of the flow might display local differences depending on the clade. However, we don’t dig into them very deeply, as results shown here imply that small variations in local morphology shouldn’t impact significantly on the total body drag of forward steady motion. Separate studies could be devoted to test if this variation in cranial morphology could affect lift production, aid manoeuvring, or impact the drag of the head’s lateral motion.

Line 123 - 124 - Another aside here, but the notably difference between ichthyosaurs and cetaceans is that parvipelvians reduce the hind limbs but not to the extent that cetaceans do. I’m curious if you tested the impact of removing just the hind limbs of the ichthyosaurs and if it made any notable difference in drag?

We have not tested this systematically for all taxa. But we have done some tests with the *Ophthalmosaurus* model (not published). Removing the hindfins in this taxon reduces the drag coefficient by about 3.5-4%. This is only a small reduction, but of course it begs the question why didn’t they lose them completely? One answer could be that hindfins are necessary for stability purposes at the cost of some extra drag. In contrast to cetaceans, in which the fluke orientation is horizontal, the fluke of ichthyosaurs is oriented vertically, and they might benefit from hindfins providing a small lift-generating surface at the rear part of the body. This interesting hypothesis could potentially be tested using stability simulations such as those used in Henderson, 2006.

Likewise, there is a degree of uncertainty regarding the upper lobe and dorsal fin proportions in most ichthyosaurs, apart from those taxa with very good soft tissue preservation, so I am curious how big of an impact the assumed form of these appendages have on drag calculations? Possibly you addressed this in your earlier work on ichthyosaurs, or in the supplement which I have not gotten to yet, so apologies if these are questions you have already tackled elsewhere.

As acknowledged by the reviewer, only few exceptional ichthyosaur specimens preserve outlines of dorsal and caudal fins. Based on these few specimens and osteological proxies (e.g. caudal vertebral morphology), in our previous work we modelled ichthyosaurs based on the most reasonable assumptions: a) pre-Jurassic ichthyosaurs were likely to not have dorsal fins or lunate flukes, b) parvipelvic ichthyosaurs were likely to have dorsal fins and lunate flukes.

We have not specifically addressed the effect of the size/shape variation of these appendages. However, flukes and dorsal fins are both swept and highly streamlined elements (as seen in modern cetaceans) and therefore would affect only minimally to the total drag of the body in forward motion. We have added a paragraph explaining these modelling assumptions and their influence in drag in the Supplementary methods section (Supplementary Data, page 17).

Line 126 - more succinct/direct phrasing would be “morphology of plesiosaurs produced higher drag than parvipelvic ichthyosaurs...” rather than the quasi double-negative here.

This has been rephrased following the reviewer’s suggestion.

Line 145-6 (OK, this partly answers my questions from line 123 above!)

That is very good to find out. The example mentioned above (*Ophthalmosaurus*) has highly reduced hindfins. The contribution of hindfins to the drag of plesiosaurs would be much larger than in ichthyosaurs, but of course they used them for propulsion. We briefly highlight in lines 145–146 that recent research indicates that the use of hindlimbs might have enhanced the propulsive efficiency in plesiosaurs and therefore compensated for the potential extra drag.

Line 221 - higher drag than other plesiosaurs of the same size right? It might be useful to specify.

Indeed, this stands when comparing animals of the same trunk dimensions. We have modified this phrase to state this more clearly (lines 245–246).

Line 253 - I buy the argument here BUT as the authors already pointed out in their introduction, comparison with living forms is hampered by the fact that there are no good extant analogues for plesiosaumorphs. I would perhaps suggest replacing ‘This is improbable because’ (which implies a degree of quantitative hypothesis testing) with ‘However,’

This has been rephrased as suggested by the reviewer (line 278).

Line 258 - 260 - Again I agree with the overall sentiment here, but I worry that the case is overstated just a bit. The authors have not done direct testing of plesiosaur swimming performance w.r.t thrust generation, rather they compared the drag produced by static models. Thus I don’t think there is strong enough evidence to ‘reject’ slow swimming speeds in plesiosaumorphs, rather they have shown that the drag produced by the plesiosaumorph body plan would not preclude them from faster swimming speeds.

So changing “rejected” to “not supported” would probably be more appropriate.

We totally agree with the reviewer and thus we have rephrased following this suggestion (line 284).

Line 303 - This pattern of increasing body size over time is typical for marine tetrapods and has been investigated extensively (e.g. Gearty et al. 2018 for a recent paper on marine mammals). It would

probably be worth discussing the results presented here within the context of that larger evolutionary pattern.

We highly appreciate this remark. Because we had centred our discussion on the hydrodynamic effects of the neck/trunk proportion we had in fact missed this very interesting point in our discussion. We have added a paragraph elaborating on this, citing various relevant references to patterns of increasing body size in secondarily aquatic tetrapods (including the one suggested by the reviewer) and also highlight the fact that this deserves further research (lines 337–341).

Line 330 - Misspelled should be “weddellonectians”

This typo has been corrected in the text (line 358).

Line 344 - Here’s a case where I think you are *understating.* Hydrodynamics almost certainly places significant constraints on neck morphology in animals that swim fully submerged. However, wording is fine as is if it is to your preference.

We thank the reviewer for making this point. We agree that we can affirm that our results do show that one of the constraints shaping the morphology of necks over evolution is the hydrodynamic constraint, and therefore we have rephrased accordingly (see line 372).

Figure 5a - very minor suggestion: inverting the order on the key will match the order in the univariate plot which would ease interpretation somewhat

We agree with this observation. The legend of this graph has been modified accordingly.

5d - The log10 plot is very difficult to read at this small scale. I suggest adjusting the elements in 5d & c so it can be enlarged (for example you don’t need to reproduce the slow to fast color scale bar 3 times & it could be moved to the lower left to make more room for the log10 plot. Alternatively it can be omitted as it is already in the supplement and does not have major impact on the key results discussed here.

This figure has been modified following the second option suggested by the reviewer. Readjusting it to improve its readability is difficult without creating a very crowded figure, therefore the small graph has been omitted and left only as supplementary material, to which we refer in the figure legend (lines 383–384).

Line 384 - ‘novaengliae’ should be ‘novaeangliae’

This typo has been corrected in this line (412) and further down the paragraph (line 416).

References (not cited in the main manuscript list)

Aoki, K. *et al.* Northern elephant seals adjust gliding and stroking patterns with changes in buoyancy: validation of at-sea metrics of body density. *Journal of Experimental Biology* **214**, 2973–2987 (2011).

Additional modifications

-We have added an explanation in figures 2 and 3 of what the error bars represent (lines 169–170; 171–173; 222–223).

-In the methods section, we added extra detail that was missing from the flow domain configuration (lines 432–434).

-Various references in the methods' section had become unlinked and were missing in the reference list. This has been corrected.

REVIEWERS' COMMENTS:

Reviewer #1 (Remarks to the Author):

The manuscript has been improved regarding the main elements of the study. It makes a definite contribution to understanding the hydrodynamics of fossil aquatic tetrapods. However, there are two concerns that need to be addressed. The first is that you have deferred to incorporate the reference of Aleyev (1977) for the drag coefficient of a plesiosaur. The rationale was that Aleyev obtained a drag coefficient from a physical model and you inferred that there were potential errors in his measurement. This may be true, but there are also potential measurements in your own methodology (i.e., water density determined by the program Fluent and known to be an underestimate of the density of seawater; animals were considered to be static without swimming motions). As you indicated, the CFD results were for relative measures to make comparisons with other animals and determine the effect of neck length of plesiosaurs. To delete Aleyev's reference would potentially imply that you were the first to report on estimates of the drag coefficients for plesiosaurs, which is patently false. The results of Aleyev can be easily included in the text after line 91, where you are reviewing other data that do not correspond to the data that you measured. You can put in a sentence indicating that a previous measurement of the drag coefficient of a plesiosaur was made that might have been an over-estimate for the reasons provided in your rebuttal. The second comment relates to from the CFD images of the dolphin and ichthyosaurs that are in Figure 1. These look identical to the images in Gutarra et al. (2019) and Gutarra and Rahman (2021). Were these images produced anew along with the new images of plesiosaurs and whales or were they simply reprinted from the previous studies. If the latter, then the references should be indicated in the figure legends of the present study.

Reviewer #2 (Remarks to the Author):

The authors have done a great job responding to reviewer comments, I apologize for my delay in the re-review.

Revisions to figure 5 benefit the overall legibility, thanks!

The new supplementary figures and tables are added in response to Reviewer 1 comments are valuable additions.

Line 20 - Minor semantic point, I suggest replacing "favouring" with "allowing" as a related constraint is different from selective pressure.

Line 28 - I appreciate the added clarification. I think "longest-persisting" might be better than "longest-lived" as that can refer to individual life-span as well. Again semantic point.

Line 29 - One more minor point: "with" is an ambiguous linking word here, is the argument that these factors (body size, metabolism & thermoregulation) are directly associated with adaptation toward a pelagic lifestyle? That seems reasonable but perhaps the phrasing in this sentence could make that link more clear, e.g., replacing "with" with "which drove the evolution of" or some equivalent phrase that suits the authors' intention.

Line 204 - The proper genus to use here would be 'Orcinus' rather than 'Orca' which is the species name.

Line 351 - It only occurred to me on the second reading but it is interesting that the first increase in neck ratio evolutionary rate near the base of Pistosauroidea is coincident with a step toward larger body size within that clade (albeit not resolved as an rate increase per fig 5d). Then a second distinct step large trunk length among Thalassophoneans is associated with reduced (or at least low overall) neck ratio. Finally, as noted by authors, an increase in trunk length among elasmosaurs is coincident with extreme neck elongation (and neck ratio evolutionary rates). Fig 5a suggests a really interesting ratchet relationship between neck ratio and trunk length. Within the scope of the paper this is covered well, but this figure reveals suggests several very interesting

hypotheses about sauropterygian evolution. Anyway this does not require revision but I hope the authors will pursue the many interesting implications of the work presented here (also noted in my previous review).

Line 367 - See semantic point from abstract above re: "favoured." There is a subtle and important distinction between so-called "enabling" and "selective" factors in evolution (see e.g. Vermeij 2016). I'd suggest "facilitated" as another clade of plesiosaurs that evolved large trunk size did not evolve long necks, suggesting the relationship is not direct.

Vermeij, G. J. (2016). Gigantism and its implications for the history of life. PLoS One, 11(1), e0146092.

RESPONSE TO REVIEWERS

REVIEWERS' COMMENTS:

Reviewer #1 (Remarks to the Author):

The manuscript has been improved regarding the main elements of the study. It makes a definite contribution to understanding the hydrodynamics of fossil aquatic tetrapods. We thank the reviewer for the positive remarks regarding our research.

However, there are two concerns that need to be addressed. The first is that you have deferred to incorporate the reference of Aleyev (1977) for the drag coefficient of a plesiosaur. The rationale was that Aleyev obtained a drag coefficient from a physical model and you inferred that there were potential errors in his measurement. This may be true, but there are also potential measurements in your own methodology (i.e., water density determined by the program Fluent and known to be an underestimate of the density of seawater; animals were considered to be static without swimming motions). As you indicated, the CFD results were for relative measures to make comparisons with other animals and determine the effect of neck length of plesiosaurs. To delete Aleyev's reference would potentially imply that you were the first to report on estimates of the drag coefficients for plesiosaurs, which is patently false. The results of Aleyev can be easily included in the text after line 91, where you are reviewing other data that do not correspond to the data that you measured. You can put in a sentence indicating that a previous measurement of the drag coefficient of a plesiosaur was made that might have been an over-estimate for the reasons provided in your rebuttal.

We thank the reviewer for this comment. As detailed in our previous response, drag coefficient is independent of the media density, and additionally, we only compare static drag coefficients. However, we strongly agree with the reviewer that not citing Aleyev's work would represent an omission of (possibly the first) calculation of the drag coefficient of plesiosaurs, and for that reason it needs to be included. We have followed the reviewer's advice, by adding a phrase relative to this study and justifying why the values are not comparable to our results (i.e. likely due to high wave drag resulting from models being very close to the air-water interface).

The second comment relates to from the CFD images of the dolphin and ichthyosaurs that are in Figure 1. These look identical to the images in Gutarra et al. (2019) and Gutarra and Rahman (2021). Were these images produced anew along with the new images of plesiosaurs and whales or were they simply reprinted from the previous studies. If the latter, then the references should be indicated in the figure legends of the present study.

We apologise for this omission. A reference has been added in the legend of figure 1, regarding the images of the bottlenose dolphin and the three ichthyosaurs, which were part of a figure in our previous study of ichthyosaurs (Gutarra et al., 2019). Additionally, we have made sure that these models, first described in Gutarra et al. 2019, are properly referenced in the methods section.

Reviewer #2 (Remarks to the Author):

The authors have done a great job responding to reviewer comments, I apologize for my delay in the re-review.

We thank the reviewer for the positive remarks, and we are very glad to hear that the changes to the manuscript are satisfactory.

Revisions to figure 5 benefit the overall legibility, thanks!

We thank the reviewer for making this suggestion in the first place.

The new supplementary figures and tables are added in response to Reviewer 1 comments are valuable additions.

These positive remarks are highly appreciated.

Line 20 - Minor semantic point, I suggest replacing “favouring” with “allowing” as a related constraint is different from selective pressure.

We agree with the suggested change. This has been rephrased accordingly.

Line 28 - I appreciate the added clarification. I think “longest-persisting” might be better than “longest-lived” as that can refer to individual life-span as well. Again semantic point.

We agree with this suggested change and thus rephrased accordingly.

Line 29 - One more minor point: “with” is an ambiguous linking word here, is the argument that these factors (body size, metabolism & thermoregulation) are directly associated with adaptation toward a pelagic lifestyle? That seems reasonable but perhaps the phrasing in this sentence could make that link more clear, e.g., replacing “with” with “which drove the evolution of” or some equivalent phrase that suits the authors’ intention.

We appreciate this remark and consider the phrase can be improved. We have established a clearer connection between the shift in habitat and body size, metabolism and thermoregulation, by replacing the word ‘with’ with ‘a shift in ecology coupled with the evolution of’.

Line 204 - The proper genus to use here would be ‘Orcinus’ rather than ‘Orca’ which is the species name.

Thank you for spotting this error. It has been corrected now and changed to the Latin name ‘Orcinus’.

Line 351 - It only occurred to me on the second reading but it is interesting that the first increase in neck ratio evolutionary rate near the base of Pistosauroida is coincident with a step toward larger body size within that clade (albeit not resolved as an rate increase per fig 5d). Then a second distinct step large trunk length among Thalassophoneans is associated with reduced (or at least low overall) neck ratio. Finally, as noted by authors, an increase in trunk length among elasmosaurs is coincident with extreme neck elongation (and neck ratio evolutionary rates). Fig 5a suggests a really interesting ratchet relationship between neck ratio and trunk length. Within the scope of the paper this is covered well, but this figure reveals suggests several very interesting hypotheses about sauropterygian evolution.

Anyway this does not require revision but I hope the authors will pursue the many interesting implications of the work presented here (also noted in my previous review). Thank you for the remark. Indeed, there are many interesting connections between these characters that are outside of the scope of this work. We expect to continue exploring the evolution of necks and the relationship between body size and neck plasticity at more basal parts of the sauropterygian tree in future work.

Line 367 - See semantic point from abstract above re: "favoured." There is a subtle and important distinction between so-called "enabling" and "selective" factors in evolution (see e.g. Vermeij 2016). I'd suggest "facilitated" as another clade of plesiosaurs that evolved large trunk size did not evolve long necks, suggesting the relationship is not direct. This is good point. Thanks also for the reference. Indeed, we agree that using 'facilitating' conveys better the sense of large trunks enabling (as opposed to conditioning) the evolution of large necks. We have changed this word as suggested.

Vermeij, G. J. (2016). Gigantism and its implications for the history of life. PLoS One, 11(1), e0146092.

Neil Kelley